# Balanced Policy Evaluation and Learning

**Nathan Kallus**
Cornell University and Cornell Tech
`kallus@cornell.edu`

## Abstract

We present a new approach to the problems of evaluating and learning personalized decision policies from observational data of past contexts, decisions, and outcomes. Only the outcome of the enacted decision is available and the historical policy is unknown. These problems arise in personalized medicine using electronic health records and in internet advertising. Existing approaches use inverse propensity weighting (or, doubly robust versions) to make historical outcome (or, residual) data look like it were generated by a new policy being evaluated or learned. But this relies on a plug-in approach that rejects data points with a decision that disagrees with the new policy, leading to high variance estimates and ineffective learning. We propose a new, balance-based approach that too makes the data look like the new policy but does so *directly* by finding weights that *optimize* for balance between the weighted data and the target policy in the given, finite sample, which is equivalent to minimizing worst-case or posterior conditional mean square error. Our policy learner proceeds as a two-level optimization problem over policies and weights. We demonstrate that this approach markedly outperforms existing ones both in evaluation and learning, which is unsurprising given the wider support of balance-based weights. We establish extensive theoretical consistency guarantees and regret bounds that support this empirical success.

## 1   Introduction

Using observational data with partially observed outcomes to develop new and effective personalized decision policies has received increased attention recently [1, 7, 8, 13, 23, 29, 41–43, 45]. The aim is to transform electronic health records to personalized treatment regimes [6], transactional records to personalized pricing strategies [5], and click- and "like"-streams to personalized advertising campaigns [8] – problems of great practical significance. Many of the existing methods rely on a reduction to weighted classification via a rejection and importance sampling technique related to inverse propensity weighting and to doubly robust estimation. However, inherent in this reduction are several shortcomings that lead to reduced personalization efficacy: it involves a naïve plug-in estimation of a denominator nuisance parameter leading either to high variance or scarcely-motivated stopgaps; it necessarily rejects a significant amount of observations leading to smaller datasets in effect; and it proceeds in a two-stage approach that is unnatural to the single learning task.

In this paper, we attempt to ameliorate these by using a new approach that directly optimizes for the balance that is achieved only on average or asymptotically by the rejection and importance sampling approach. We demonstrate that this new approach provides improved performance and explain why. And, we provide extensive theory to characterize the behavior of the new methods. The proofs are given in the supplementary material.

### 1.1   Setting, Notation, and Problem Description

The problem we consider is how to choose the best of $m$ treatments based on an observation of covariates $x \in \mathcal{X} \subseteq \mathbb{R}^d$ (also known as a context). An instance is characterized by the random variables $X \in \mathcal{X}$ and $Y(1), \dots, Y(m) \in \mathbb{R}$, where $X$ denotes the covariates and $Y(t)$ for $t \in [m] =$

$\{1, \ldots, m\}$ is the outcome that would be derived from applying treatment $t$. We always assume that *smaller* outcomes are preferable, *i.e.*, $Y(t)$ corresponds to costs or *negative* rewards.

A policy is a map $\pi : \mathcal{X} \to \Delta^m$ from observations of covariates to a probability vector in the $m$-simplex $\Delta^m = \{p \in \mathbb{R}_+^m : \sum_{t=1}^m p_t = 1\}$. Given an observation of covariates $x$, the policy $\pi$ specifies that treatment $t$ should be applied with probability $\pi_t(x)$. There are two key tasks of interest: policy evaluation and policy learning. In policy evaluation, we wish to evaluate the performance of a given policy based on historical data. This is also known as *off*-policy evaluation, highlighting the fact that the historical data was not necessarily generated by the policy in question. In policy learning, we wish to determine a policy that has good performance.

We consider doing both tasks based on data consisting of $n$ *passive, historical observations* of covariate, treatment, and outcome: $S_n = \{(X_1, T_1, Y_1), \ldots, (X_n, T_n, Y_n)\}$, where the observed outcome $Y_i = Y_i(T_i)$ corresponds *only* to the treatment $T_i$ historically applied. We use the notation $X_{1:n}$ to denote the data tuple $(X_1, \ldots, X_n)$. The data is assumed to be iid. That is, the data is generated by drawing from a stationary population of instances $(X, T, Y(1), \ldots, Y(m))$ and observing a censored form of this draw given by $(X, T, Y(T))$.[1] From the (unknown) joint distribution of $(X, T)$ in the population, we define the (unknown) propensity function $\varphi_t(x) = \mathbb{P}(T = t \mid X = x) = \mathbb{E}[\delta_{Tt} \mid X = x]$, where $\delta_{st} = \mathbb{I}[s = t]$ is the Kronecker delta. And, from the (unknown) joint distribution of $(X, Y(t))$ in the population, we define the (unknown) mean-outcome function $\mu_t(x) = \mathbb{E}[Y(t) \mid X = x]$. We use the notation $\varphi(x) = (\varphi_1(x), \ldots, \varphi_m(x))$ and $\mu(x) = (\mu_1(x), \ldots, \mu_m(x))$.

Apart from being iid, we also assume the data satisfies unconfoundedness:

**Assumption 1.** For each $t \in [m]$: $Y(t)$ is independent of $T$ given $X$, *i.e.*, $Y(t) \perp\!\!\!\perp T \mid X$.

This assumption is equivalent to there being a *logging policy* $\varphi$ that generated the data by prescribing treatment $t$ with probability $\varphi_t(X_i)$ to each instance $i$ and recording the outcome $Y_i = Y_i(T_i)$. Therefore, especially in the case where the logging policy $\varphi_t$ is in fact *known* to the user, the problem is often called learning from logged bandit feedback [41, 42].

In policy evaluation, given a policy $\pi$, we wish to estimate its *sample-average policy effect* (SAPE),
$$\text{SAPE}(\pi) = \tfrac{1}{n} \sum_{i=1}^n \sum_{t=1}^m \pi_t(X_i)\mu_t(X_i),$$
by an estimator $\hat{\tau}(\pi) = \hat{\tau}(\pi; X_{1:n}, T_{1:n}, Y_{1:n})$ that depends only on the observed data and the policy $\pi$. The SAPE quantifies the average outcome that a policy $\pi$ would induce in the sample and hence measures its risk. SAPE is strongly consistent for the *population-average policy effect* (PAPE):
$$\text{PAPE}(\pi) = \mathbb{E}[\text{SAPE}(\pi)] = \mathbb{E}[\sum_{t=1}^m \pi_t(X)\mu_t(X)] = \mathbb{E}[Y(\tilde{T}_{\pi(X)})],$$
where $\tilde{T}_{\pi(x)}$ is defined as $\pi$'s random draw of treatment when $X = x$, $\tilde{T}_{\pi(x)} \sim \text{Multinomial}(\pi(x))$. Moreover, if $\pi^*$ is such that $\pi_t^*(x) > 0 \iff t \in \text{argmin}_{s \in [m]} \mu_s(x)$, then $\widehat{R}(\pi) = \text{SAPE}(\pi) - \text{SAPE}(\pi^*)$ is the *regret* of $\pi$ [10]. The policy evaluation task is closely related to causal effect estimation [19] where, for $m = 2$, one is interested in estimating the sample and population average treatment effects: $\text{SATE} = \tfrac{1}{n}\sum_{i=1}^n(\mu_2(X_i) - \mu_1(X_i))$, $\text{PATE} = \mathbb{E}[\text{SATE}] = \mathbb{E}[Y(2) - Y(1)]$.

In policy learning, we wish to find a policy $\hat{\pi}$ that achieves small outcomes, *i.e.*, small SAPE and PAPE. The optimal policy $\pi^*$ minimizes both $\text{SAPE}(\pi)$ and $\text{PAPE}(\pi)$ over all functions $\mathcal{X} \to \Delta^m$.

## 1.2 Existing Approaches and Related Work

The so-called "direct" approach fits regression estimates $\hat{\mu}_t$ of $\mu_t$ on each dataset $\{(X_i, Y_i) : T_i = t\}$, $t \in [m]$. Given these estimates, it estimates SAPE in a *plug-in* fashion:
$$\hat{\tau}^{\text{direct}}(\pi) = \tfrac{1}{n} \sum_{i=1}^n \sum_{t=1}^m \pi_t(X_i)\hat{\mu}_t(X_i).$$
A policy is learned either by $\hat{\pi}^{\text{direct}}(x) = \text{argmin}_{t \in [m]} \hat{\mu}_t(x)$ or by minimizing $\hat{\tau}^{\text{direct}}(\pi)$ over $\pi \in \Pi$ [33]. However, direct approaches may not generalize as well as weighting-based approaches [7].

Weighting-based approaches seek weights based on covariate and treatment data $W(\pi) = W(\pi; X_{1:n}, T_{1:n})$ that make the outcome data, when reweighted, look as though it were generated by the policy being evaluated or learned, giving rise to estimators that have the form
$$\hat{\tau}_W = \tfrac{1}{n} \sum_{i=1}^n W_i Y_i.$$

Bottou et al. [8], *e.g.*, propose to use inverse propensity weighting (IPW). Noting that [17, 18] $\mathrm{SAPE}(\pi) = \mathbb{E}[\frac{1}{n} \sum_{i=1}^{n} Y_i \times \pi_{T_i}(X_i)/\varphi_{T_i}(X_i) \mid X_{1:n}]$, one first fits a probabilistic classification model $\hat{\varphi}$ to $\{(X_i, T_i) : i \in [n]\}$ and then estimates SAPE in an alternate but also *plug-in* fashion:

$$\hat{\tau}^{\mathrm{IPW}}(\pi) = \hat{\tau}_{W^{\mathrm{IPW}}(\pi)}, \quad W_i^{\mathrm{IPW}}(\pi) = \pi_{T_i}(X_i)/\hat{\varphi}_{T_i}(X_i)$$

For a deterministic policy, $\pi_t(x) \in \{0, 1\}$, this can be interpreted as a rejection and importance sampling approach [29, 41]: reject samples where the observed treatment does not match $\pi$'s recommendation and up-weight those that do by the inverse (estimated) propensity. For deterministic policies $\pi_t(x) \in \{0, 1\}$, we have that $\pi_T(X) = \delta_{T, \tilde{T}_{\pi(X)}}$ is the complement of 0-1 loss of $\pi(X)$ in predicting $T$. By scaling and constant shifts, one can therefore reduce minimizing $\hat{\tau}^{\mathrm{IPW}}(\pi)$ over policies $\pi \in \Pi$ to minimizing a *weighted classification* loss over *classifiers* $\pi \in \Pi$, providing a reduction to weighted classification [7, 45].

Given both $\hat{\mu}(x)$ and $\hat{\varphi}(x)$ estimates, Dudík et al. [13] propose a weighting-based approach that combines the direct and IPW approaches by adapting the doubly robust (DR) estimator [11, 34, 35, 38]:

$$\hat{\tau}^{\mathrm{DR}}(\pi) = \frac{1}{n} \sum_{i=1}^{n} \sum_{t=1}^{m} \pi_t(X_i)\hat{\mu}_t(X_i) + \frac{1}{n} \sum_{i=1}^{n} (Y_i - \hat{\mu}_{T_i}(X_i))\pi_{T_i}(X_i)/\hat{\varphi}_{T_i}(X_i).$$

$\hat{\tau}^{\mathrm{DR}}(\pi)$ can be understood either as *debiasing* the direct estimator by via the reweighted residuals $\hat{\epsilon}_i = Y_i - \hat{\mu}_{T_i}(X_i)$ or as *denoising* the IPW estimator by subtracting the conditional mean from $Y_i$. As its bias is multiplicative in the biases of the regression and propensity estimates, the estimator is consistent so long as one of the estimates is consistent. For policy learning, [1, 13] minimize this estimator via weighted classification. Athey and Wager [1] provide a tight and favorable analysis of the corresponding uniform consistency (and hence regret) of the DR approach to policy learning.

Based on the fact that $1 = \mathbb{E}[\pi_T(X)/\varphi_T(X)]$, a normalized IPW (NIPW) estimator is given by normalizing the weights so they sum to $n$, a common practice in causal effect estimation [2, 31]:

$$\hat{\tau}^{\mathrm{NIPW}}(\pi) = \hat{\tau}_{W^{\mathrm{NIPW}}(\pi)}, \quad W_i^{\mathrm{NIPW}}(\pi) = W_i^{\mathrm{IPW}}(\pi)/\sum_{i'=1}^{n} W_{i'}^{\mathrm{IPW}}(\pi).$$

Any IPW approaches are subject to considerable variance because the plugged-in propensities are in the denominator so that small errors can have *outsize* effects on the total estimate. Another stopgap measure is to *clip* the propensities [14, 20] resulting in the clipped IPW (CIPW) estimator:

$$\hat{\tau}^{M\text{-CIPW}}(\pi) = \hat{\tau}_{W^{M\text{-CIPW}}(\pi)}, \quad W_i^{M\text{-CIPW}}(\pi) = \pi_{T_i}(X_i)/\max\{M, \hat{\varphi}_{T_i}(X_i)\}.$$

While effective in reducing variance, the practice remains ad-hoc, loses the unbiasedness of IPW (with true propensities), and requires the tuning of $M$. For policy learning, Swaminathan and Joachims [42] propose to minimizes over $\pi \in \Pi$ the $M$-CIPW estimator plus a regularization term of the sample variance of the estimator, which they term POEM. The sample variance scales with the level of overlap between $\pi$ and $T_{1:n}$, *i.e.*, the prevalence of $\pi_{T_i}(X_i) > 0$. Indeed, when the policy class $\Pi$ is very flexible relative to $n$ and if outcomes are nonnegative, then the *anti*-logging policy $\pi_{T_i}(X_i) = 0$ minimizes any of the above estimates. POEM avoids learning the anti-logging policy by regularizing overlap, reducing variance but limiting novelty of $\pi$. A refinement, SNPOEM [43] uses a normalized *and* clipped IPW (NCIPW) estimator (and regularizes variance):

$$\hat{\tau}^{M\text{-NCIPW}}(\pi) = \hat{\tau}_{W^{M\text{-NCIPW}}(\pi)}, \quad W_i^{M\text{-NCIPW}}(\pi) = W_i^{M\text{-CIPW}}(\pi)/\sum_{i'=1}^{n} W_{i'}^{M\text{-CIPW}}(\pi).$$

Kallus and Zhou [26] generalize the IPW approach to a continuum of treatments. Kallus and Zhou [25] suggest a minimax approach to perturbations of the weights to account for confounding factors. Kallus [23] proposes a recursive partitioning approach to policy learning, the Personalization Tree (PT) and Personalization Forest (PF), that dynamically learns both weights and policy, but still uses within-partition IPW with dynamically estimated propensities.

### 1.3 A Balance-Based Approach

**Shortcomings in existing approaches.** All of the above weighting-based approaches seek to reweight the historical data so that they look as though they were generated by the policy being evaluated or learned. Similarly, the DR approach seeks to make the historical *residuals* look like those that would be generated under the policy in question so to remove bias from the estimated regression model of the direct approach. However, the way these methods achieve this through various forms and versions of inverse propensity weighting, has three critical shortcomings:

(1) By taking a simple plug-in approach for a nuisance parameter (propensities) that appears in the *denominator*, existing weighting-based methods are either subject to *very high* variance or must rely on scarcely-motivated stopgap measures such as clipping (see also [27]).

(2) In the case of deterministic policies (such as an optimal policy), existing methods all have weights that are multiples of $\pi_{T_i}(X_i)$, which means that one necessarily *throws away* every data point $T_i$ that does not agree with the new policy recommendation $\tilde{T}_{\pi(X_i)}$. This means that one is essentially only using a much smaller dataset than is available, leading again to higher variance.[2]

(3) The existing weighting-based methods all proceed in two stages: first estimate propensities and then plug these in to a derived estimator (when the logging policy is unknown). On the one hand, this raises model specification concerns, and on the other, is unsatisfactory when the task at hand is not inherently two-staged – we wish *only* to evaluate or learn policies, not to learn propensities.

**A new approach.** We propose a balance-based approach that, like the existing weighting-based methods, also reweights the historical data to make it look as though they were generated by the policy being evaluated or learned and potentially denoises outcomes in a doubly robust fashion, *but rather* than doing so circuitously via a plug-in approach, we do it *directly* by finding weights that *optimize* for *balance* between the weighted data and the target policy in the given, finite sample.

In particular, we formalize balance as a discrepancy between the reweighted historical covariate distribution and that induced by the target policy and prove that it is directly related to the worst-case *conditional mean square error* (CMSE) of *any* weighting-based estimator. Given a policy $\pi$, we then propose to choose (policy-dependent) weights $W^*(\pi)$ that *optimize* the worst-case CMSE and therefore achieve excellent balance while controlling for variance. For evaluation, we use these optimal weights to evaluate the performance of $\pi$ by the estimator $\hat{\tau}_{W^*(\pi)}$ as well as a doubly robust version. For learning, we propose a *bilevel* optimization problem: minimize over $\pi \in \Pi$, the estimated risk $\hat{\tau}_{W^*(\pi)}$ (or a doubly robust version thereof and potentially plus a regularization term), given by the weights $W^*(\pi)$ that minimize the estimation error. Our empirical results show the stark benefit of this approach while our main theoretical results (Thm. 6, Cor. 7) establish vanishing regret bounds.

## 2 Balanced Evaluation

### 2.1 CMSE and Worst-Case CMSE

We begin by presenting the approach in the context of evaluation. Given a policy $\pi$, consider *any* weights $W = W(\pi; X_{1:n}, T_{1:n})$ that are based on the covariate and treatment data. Given these weights we can consider both a simple weighted estimator as well as a $W$-weighted doubly robust estimator given a regression estimate $\hat{\mu}$:

$$\hat{\tau}_W = \frac{1}{n} \sum_{i=1}^{n} W_i Y_i, \quad \hat{\tau}_{W,\hat{\mu}} = \frac{1}{n} \sum_{i=1}^{n} \sum_{t=1}^{m} \pi_t(X_i) \hat{\mu}_t(X_i) + \frac{1}{n} \sum_{i=1}^{n} W_i(Y_i - \hat{\mu}_{T_i}(X_i)).$$

We can measure the risk of either such estimator as the conditional mean square error (CMSE), conditioned on all of the data upon which the chosen weights depend:

$$\text{CMSE}(\hat{\tau}, \pi) = \mathbb{E}[(\hat{\tau} - \text{SAPE}(\pi))^2 \mid X_{1:n}, T_{1:n}].$$

Minimal CMSE is the target of choosing weights for weighting-based policy evaluation. Basic manipulations under the unconfoundedness assumption decompose the CMSE of any weighting-based policy evaluation estimator into its conditional bias and variance:

**Theorem 1.** *Let $\epsilon_i = Y_i - \mu_{T_i}(X_i)$ and $\Sigma = \text{diag}(\mathbb{E}[\epsilon_1^2 \mid X_1, T_1], \ldots, \mathbb{E}[\epsilon_n^2 \mid X_n, T_n])$. Define*

$$B_t(W, \pi_t; f_t) = \frac{1}{n} \sum_{i=1}^{n} (W_i \delta_{T_i t} - \pi_t(X_i)) f_t(X_i) \quad \text{and} \quad B(W, \pi; f) = \sum_{t=1}^{m} B_t(W, \pi_t; f_t)$$

*Then we have that:* $\qquad\qquad\qquad\qquad\qquad \hat{\tau}_W - \text{SAPE}(\pi) = B(W, \pi; \mu) + \frac{1}{n} \sum_{i=1}^{n} W_i \epsilon_i.$

*Moreover, under Asn. 1:* $\qquad\qquad\qquad\qquad \text{CMSE}(\hat{\tau}_W, \pi) = B^2(W, \pi; \mu) + \frac{1}{n^2} W^T \Sigma W.$

Figure 1: The setting in Ex. 1

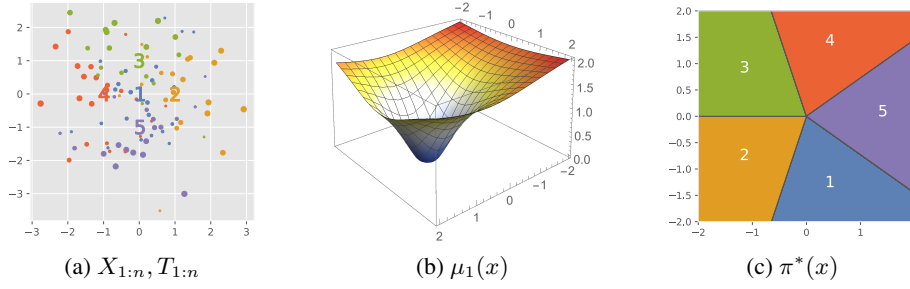

(a) $X_{1:n}, T_{1:n}$        (b) $\mu_1(x)$        (c) $\pi^*(x)$

Table 1: Policy evaluation performance in Ex. 1

| Weights $W$ | Vanilla $\hat{\tau}_W$ | | | Doubly robust $\hat{\tau}_{W,\hat{\mu}}$ | | | $\|W\|_0$ |
|---|---|---|---|---|---|---|---|
| | RMSE | Bias | SD | RMSE | Bias | SD | |
| IPW, $\varphi$ | 2.209 | $-0.005$ | 2.209 | 4.196 | 0.435 | 4.174 | $13.6 \pm 2.9$ |
| IPW, $\hat{\varphi}$ | 0.568 | $-0.514$ | 0.242 | 0.428 | 0.230 | 0.361 | $13.6 \pm 2.9$ |
| .05-CIPW, $\varphi$ | 0.581 | $-0.491$ | 0.310 | 0.520 | 0.259 | 0.451 | $13.6 \pm 2.9$ |
| .05-CIPW, $\hat{\varphi}$ | 0.568 | $-0.514$ | 0.242 | 0.428 | 0.230 | 0.361 | $13.6 \pm 2.9$ |
| NIPW, $\varphi$ | 0.519 | $-0.181$ | 0.487 | 0.754 | 0.408 | 0.634 | $13.6 \pm 2.9$ |
| NIPW, $\hat{\varphi}$ | 0.463 | $-0.251$ | 0.390 | 0.692 | 0.467 | 0.511 | $13.6 \pm 2.9$ |
| .05-NCIPW, $\varphi$ | 0.485 | $-0.250$ | 0.415 | 0.724 | 0.471 | 0.550 | $13.6 \pm 2.9$ |
| .05-NCIPW, $\hat{\varphi}$ | 0.463 | $-0.251$ | 0.390 | 0.692 | 0.467 | 0.511 | $13.6 \pm 2.9$ |
| Balanced eval | **0.280** | 0.227 | 0.163 | **0.251** | $-0.006$ | 0.251 | $90.7 \pm 3.2$ |

**Corollary 2.** *Let $\hat{\mu}$ be given such that $\hat{\mu} \perp\!\!\!\perp Y_{1:n} \mid X_{1:n}, T_{1:n}$ (e.g., trained on a split sample). Then we have that:* $\hat{\tau}_{W,\hat{\mu}} - \mathrm{SAPE}(\pi) = B(W, \pi; \mu - \hat{\mu}) + \frac{1}{n}\sum_{i=1}^n W_i \epsilon_i$. *Moreover, under Asn. 1:* $\mathrm{CMSE}(\hat{\tau}_{W,\hat{\mu}}, \pi) = B^2(W, \pi; \mu - \hat{\mu}) + \frac{1}{n^2}W^T \Sigma W$.

In Thm. 1 and Cor. 2, $B(W, \pi; \mu)$ and $B(W, \pi; \mu - \hat{\mu})$ are precisely the conditional *bias* in evaluating $\pi$ for $\hat{\tau}_W$ and $\hat{\tau}_{W,\hat{\mu}}$, respectively, and $\frac{1}{n^2}W^T\Sigma W$ the conditional *variance* for both. In particular, $B_t(W, \pi_t; \mu_t)$ or $B_t(W, \pi_t; \mu_t - \hat{\mu}_t)$ is the conditional bias in evaluating the effect on the instances where $\pi$ assigns $t$. Note that for any function $f_t$, $B_t(W, \pi_t; f_t)$ corresponds to the *discrepancy* between the $f_t(X)$-moments of the measure $\nu_{t,\pi}(A) = \frac{1}{n}\sum_{i=1}^n \pi_t(X_i)\mathbb{I}\,[X_i \in A]$ on $\mathcal{X}$ and the measure $\nu_{t,W}(A) = \frac{1}{n}\sum_{i=1}^n W_i\delta_{T_it}\mathbb{I}\,[X_i \in A]$. The sum $B(W, \pi; f)$ corresponds to the sum of moment discrepancies over the components of $f = (f_1, \ldots, f_m)$ between these measures. The moment discrepancy of interest is that of $f = \mu$ or $f = \mu - \hat{\mu}$, but neither of these are known.

Balanced policy evaluation seeks weights $W$ to minimize a combination of *imbalance*, given by the worst-case value of $B(W, \pi; f)$ over functions $f$, and *variance*, given by the norm of weights $W^T\Lambda W$ for a specified positive semidefinite (PSD) matrix $\Lambda$. This follows a general approach introduced by [22, 24] of finding optimal balancing weights that optimize a given CMSE objective directly rather than via a plug-in approach. Any choice of $\|\cdot\|$ gives rise to a *worst-case CMSE* objective for policy evaluation:

$$\mathfrak{E}^2(W, \pi; \|\cdot\|, \Lambda) = \sup_{\|f\| \le 1} B^2(W, \pi; f) + \frac{1}{n^2}W^T\Lambda W.$$

Here, we focus on $\|\cdot\|$ given by the *direct product of reproducing kernel Hilbert spaces (RKHS)*:

$$\|f\|_{p, \mathcal{K}_{1:m}, \gamma_{1:m}} = \left(\sum_{t=1}^m \|f_t\|_{\mathcal{K}_t}^p / \gamma_t^p\right)^{1/p},$$

where $\|\cdot\|_{\mathcal{K}_t}$ is the norm of the RKHS given by the PSD kernel $\mathcal{K}_t(\cdot, \cdot) : \mathcal{X}^2 \to \mathbb{R}$, *i.e.*, the unique completion of $\mathrm{span}(\mathcal{K}_t(x, \cdot) : x \in \mathcal{X})$ endowed with $\langle \mathcal{K}_t(x, \cdot), \mathcal{K}_t(x', \cdot)\rangle = \mathcal{K}_t(x, x')$ [see 39]. We say $\|f\|_{\mathcal{K}_t} = \infty$ if $f$ is not in the RKHS. One example of a kernel is the Mahalanobis RBF kernel: $\mathcal{K}_s(x, x') = \exp(-(x - x')^T \hat{S}^{-1}(x - x')/s^2)$ where $\hat{S}$ is the sample covariance of $X_{1:n}$ and $s$ is a parameter. For such an RKHS product norm, we can decompose the worst-case objective into the discrepancies in each treatment as well as characterize it as a posterior (rather than worst-case) risk.

**Lemma 1.** *Let $\mathfrak{B}_t^2(W, \pi_t; \|\cdot\|_{\mathcal{K}_t}) = \sum_{i,j=1}^n (W_i\delta_{T_it} - \pi_t(X_i))(W_j\delta_{T_jt} - \pi_t(X_j))\mathcal{K}_t(X_i, X_j)$ and $1/p + 1/q = 1$. Then*

$$\mathfrak{E}^2(W, \pi; \|\cdot\|_{p, \mathcal{K}_{1:m}, \gamma_{1:m}}, \Lambda) = \left(\sum_{t=1}^m \gamma_t^q \mathfrak{B}_t^q(W, \pi_t; \|\cdot\|_{\mathcal{K}_t})\right)^{2/q} + \frac{1}{n^2}W^T\Lambda W.$$

*Moreover, if $p = 2$ and $\mu_t$ has a Gaussian process prior [44] with mean $f_t$ and covariance $\gamma_t \mathcal{K}_t$ then*

$$\text{CMSE}(\hat{\tau}_{W,f}, \pi) = \mathfrak{E}^2(W, \pi; \|\cdot\|_{p,\mathcal{K}_{1:m},\gamma_{1:m}}, \Sigma),$$

*where the CMSE marginalizes over $\mu$. This gives the CMSE of $\hat{\tau}_W$ for $f$ constant or $\hat{\tau}_{W,\hat{\mu}}$ for $f = \hat{\mu}$.*

The second statement in Lemma 1 suggests that, in practice, model selection of $\gamma_{1:m}$, $\Lambda$, and kernel hyperparameters such as $s$ or even $\hat{S}$, can done by the marginal likelihood method [see 44, Ch. 5].

## 2.2 Evaluation Using Optimal Balancing Weights

Our policy evaluation estimates are given by either the estimator $\hat{\tau}_{W^*(\pi;\|\cdot\|,\Lambda)}$ or $\hat{\tau}_{W^*(\pi;\|\cdot\|,\Lambda),\hat{\mu}}$ where $W^*(\pi) = W^*(\pi; \|\cdot\|, \Lambda)$ is the *minimizer* of $\mathfrak{E}^2(W, \pi; \|\cdot\|, \Lambda)$ over the space of all weights $W$ that sum to $n$, $\mathcal{W} = \{W \in \mathbb{R}_+^n : \sum_{i=1}^n W_i = n\} = n\Delta^n$. Specifically,

$$W^*(\pi; \|\cdot\|, \Lambda) \in \operatorname{argmin}_{W \in \mathcal{W}} \mathfrak{E}^2(W, \pi; \|\cdot\|, \Lambda).$$

When $\|\cdot\| = \|\cdot\|_{p,\mathcal{K}_{1:m},\gamma_{1:m}}$, this problem is a quadratic program for $p = 2$ and a second-order cone program for $p = 1, \infty$. Both are efficiently solvable [9]. In practice, we solve these using Gurobi 7.0.

In Lemma 1, $\mathfrak{B}_t(W, \pi_t; \|\cdot\|_{\mathcal{K}_t})$ measures the *imbalance* between $\nu_{t,\pi}$ and $\nu_{t,W}$ as the *worst-case* discrepancy in means over functions in the unit ball of an RKHS. In fact, as a distributional distance metric, it is the maximum mean discrepancy (MMD) used, for example, for testing whether two samples come from the same distribution [16]. Thus, minimizing $\mathfrak{E}^2(W, \pi; \|\cdot\|_{p,\mathcal{K}_{1:m},\gamma_{1:m}}, \Lambda)$ is simply seeking the weights $W$ that balance $\nu_{t,\pi}$ and $\nu_{t,W}$ subject to variance regularization in $W$.

**Example 1.** We demonstrate balanced evaluation with a mixture of $m = 5$ Gaussians: $X \mid T \sim \mathcal{N}(\overline{X}_T, I_{2\times 2})$, $\overline{X}_1 = (0,0)$, $\overline{X}_t = (\text{Re}, \text{Im})(e^{i2\pi(t-2)/(m-1)})$ for $t = 2, \ldots, m$, and $T \sim \text{Multinomial}(1/5, \ldots, 1/5)$. Fix a draw of $X_{1:n}, T_{1:n}$ with $n = 100$ shown in Fig. 1a (numpy seed 0). Color denotes $T_i$ and size denotes $\varphi_{T_i}(X_i)$. The centers $\overline{X}_t$ are marked by a colored number. Next, we let $\mu_t(x) = \exp(1 - 1/\|x - \chi_t\|_2)$ where $\chi_t = (\text{Re}, \text{Im})(e^{-i2\pi t/m}/\sqrt{2})$ for $t \in [m]$, $\epsilon_i \sim \mathcal{N}(0, \sigma)$, and $\sigma = 1$. Fig. 1b plots $\mu_1(x)$. Fig. 1c shows the corresponding optimal policy $\pi^*$.

Next we consider evaluating $\pi^*$. Fixing $X_{1:n}$ as in Fig. 1a, we have $\text{SAPE}(\pi^*) = 0.852$. With $X_{1:n}$ fixed, we draw 1000 replications of $T_{1:n}, Y_{1:n}$ from their conditional distribution. For each replication, we fit $\hat{\varphi}$ by estimating the (well-specified) Gaussian mixture by maximum likelihood and fit $\hat{\mu}$ using $m$ separate gradient-boosted tree models (sklearn defaults). We consider evaluating $\pi^*$ either using the vanilla estimator $\hat{\tau}_W$ or the doubly robust estimator $\hat{\tau}_{W,\hat{\mu}}$ for $W$ either chosen in the 4 different standard ways laid out in Sec. 1.2, using either the true $\varphi$ or the estimated $\hat{\varphi}$, or chosen by the balanced evaluation approach using untuned parameters (rather than fit by marginal likelihood) using the standard ($s = 1$) Mahalanobis RBF kernel for $\mathcal{K}_t$, $\|f\|^2 = \sum_{t=1}^m \|f_t\|_{\mathcal{K}_t}^2$, and $\Lambda = I$. (Note that this *misspecifies* the outcome model, $\|\mu_t\|_{\mathcal{K}_t} = \infty$.) We tabulate the results in Tab. 1.

We note a few observations on the standard approaches: vanilla IPW with true $\varphi$ has zero bias but large SD (standard deviation) and hence RMSE (root mean square error); a DR approach improves on a vanilla IPW with $\hat{\varphi}$ by reducing bias; clipping and normalizing IPW reduces SD. The balanced evaluation approach achieves the best RMSE by a clear margin, with the vanilla estimator beating all standard vanilla *and* DR estimators and the DR estimator providing a further improvement by nearly eliminating bias (but increasing SD). The marked success of the balanced approach is *unsurprising* when considering the support $\|W\|_0 = \sum_{i=1}^n \mathbb{I}[W_i > 0]$ of the weights. All standard approaches use weights that are multiples of $\pi_{T_i}(X_i)$, limiting support to the overlap between $\pi$ and $T_{1:n}$, which hovers around 10–16 over replications. The balanced approach uses weights that have significantly wider support, around 88–94. In light of this, the success of the balanced approach is expected.

## 2.3 Consistent Evaluation

Next we consider the question of consistent evaluation: under what conditions can we guarantee that $\hat{\tau}_{W^*(\pi)} - \text{SAPE}(\pi)$ and $\hat{\tau}_{W^*(\pi),\hat{\mu}} - \text{SAPE}(\pi)$ converge to zero and at what rates.

One key requirement for consistent evaluation is a weak form of overlap between the historical data and the target policy to be evaluated using this data:

**Assumption 2** (Weak overlap). $\mathbb{P}(\varphi_t(X) > 0 \vee \pi_t(X) = 0) = 1 \ \forall t \in [m]$, $\mathbb{E}[\pi_T^2(X)/\varphi_T^2(X)] < \infty$.

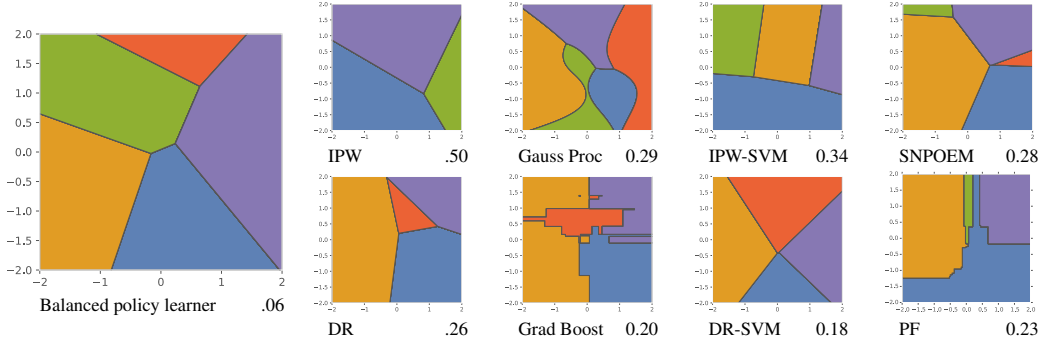

Figure 2: Policy learning results in Ex. 2; numbers denote regret

This ensures that if $\pi$ can assign treatment $t$ to $X$ then the data will have some examples of units with similar covariates being given treatment $t$; otherwise, we can never say what the outcome might look like. Another key requirement is specification. If the mean-outcome function is well-specified in that it is in the RKHS product used to compute $W^*(\pi)$ then convergence at rate $1/\sqrt{n}$ is guaranteed. Otherwise, for a doubly robust estimator, if the regression estimate is well-specified then consistency is still guaranteed. In lieu of specification, consistency is also guaranteed if the RKHS product consists of $C_0$-universal kernels, defined below, such as the RBF kernel [40].

**Definition 1.** A PSD kernel $\mathcal{K}$ on a Hausdorff $\mathcal{X}$ (*e.g.*, $\mathbb{R}^d$) is $C_0$-*universal* if, for any continuous function $g : \mathcal{X} \to \mathbb{R}$ with compact support (*i.e.*, for some $C$ compact, $\{x : g(x) \neq 0\} \subseteq C$) and $\eta > 0$, there exists $m, \alpha_1, x_1, \ldots, \alpha_m, x_m$ such that $\sup_{x \in \mathcal{X}} | \sum_{j=1}^{m} \alpha_i \mathcal{K}(x_j, x) - g(x)| \leq \eta$.

**Theorem 3.** *Fix $\pi$ and let $W_n^*(\pi) = W_n^*(\pi; \|f\|_{p, \mathcal{K}_{1:m}, \gamma_{n,1:m}}, \Lambda_n)$ with $0 \prec \underline{\kappa}I \preceq \Lambda_n \preceq \overline{\kappa}I$, $0 < \underline{\gamma} \leq \gamma_{n,t} \leq \overline{\gamma} \, \forall t \in [m]$ for each $n$. Suppose Asns. 1 and 2 hold,* $\mathrm{Var}(Y \mid X)$ *a.s. bounded,* $\mathbb{E}[\sqrt{\mathcal{K}_t(X, X)}] < \infty$, *and* $\mathbb{E}[\mathcal{K}_t(X, X)\pi_T^2(X)/\varphi_T^2(X)] < \infty$. *Then the following two results hold:*
(a) *If $\|\mu_t\|_{\mathcal{K}_t} < \infty$ for all $t \in [m]$:* $\qquad\qquad\qquad \hat{\tau}_{W_n^*(\pi)} - \mathrm{SAPE}(\pi) = O_p(1/\sqrt{n})$.
(b) *If $\mathcal{K}_t$ is $C_0$-universal for all $t \in [m]$:* $\qquad\qquad\quad \hat{\tau}_{W_n^*(\pi)} - \mathrm{SAPE}(\pi) = o_p(1)$.

The key assumptions of Thm. 3 are unconfoundedness, overlap, and bounded variance. The other conditions simply guide the choice of method parameters. The two conditions on the kernel are trivial for bounded kernels like the RBF kernel. An analogous result for the DR estimator is a corollary.

**Corollary 4.** *Suppose the assumptions of Thm. 3 hold and. Then*
(a) *If $\|\hat{\mu}_{nt} - \mu_t\|_{\mathcal{K}_t} = o_p(1) \, \forall t \in [m]$:*
$$\hat{\tau}_{W_n^*(\pi), \hat{\mu}_n} - \mathrm{SAPE}(\pi) = (\tfrac{1}{n^2} \textstyle\sum_{i=1}^{n} W_{ni}^{*\,2} \, \mathrm{Var}(Y_i \mid X_i))^{1/2} + o_p(1/\sqrt{n}).$$
(b) *If $\|\hat{\mu}_n(X) - \mu(X)\|_2 = O_p(r(n))$, $r(n) = \Omega(1/\sqrt{n})$:* $\quad \hat{\tau}_{W_n^*(\pi), \hat{\mu}_n} - \mathrm{SAPE}(\pi) = O_p(r(n))$.
(c) *If $\|\mu_t\|_{\mathcal{K}_t} < \infty, \|\hat{\mu}_{nt}\|_{\mathcal{K}_t} = O_p(1)$ for all $t \in [m]$:* $\quad \hat{\tau}_{W_n^*(\pi), \hat{\mu}_n} - \mathrm{SAPE}(\pi) = O_p(1/\sqrt{n})$.
(d) *If $\mathcal{K}_t$ is $C_0$-universal for all $t \in [m]$:* $\qquad\qquad\qquad \hat{\tau}_{W_n^*(\pi), \hat{\mu}_n} - \mathrm{SAPE}(\pi) = o_p(1)$.

Cor. 4(a) is the case where both the balancing weights and the regression function are well-specified, in which case the multiplicative bias disappears faster than $o_p(1/\sqrt{n})$, leaving us only with the irreducible residual variance, leading to an *efficient* evaluation. The other cases concern the "doubly robust" nature of the balanced DR estimator: Cor. 4(b) requires only that the regression be consistent and Cor. 4(c)-(d) require only the balancing weights to be consistent.

## 3 Balanced Learning

Next we consider a balanced approach to policy learning. Given a policy class $\Pi \subset [\mathcal{X} \to \Delta^m]$, we let the **balanced policy learner** yield the policy $\pi \in \Pi$ that minimizes the balanced policy evaluation using either a vanilla or DR estimator plus a potential regularization term in the worst-case/posterior CMSE of the evaluation. We formulate this as a *bilevel* optimization problem:

$$\hat{\pi}^{\mathrm{bal}} \in \mathrm{argmin}_\pi \{ \hat{\tau}_W + \lambda \mathfrak{E}(W, \pi; \|\cdot\|, \Lambda) : \pi \in \Pi, W \in \mathrm{argmin}_{W \in \mathcal{W}} \, \mathfrak{E}^2(W, \pi; \|\cdot\|, \Lambda) \} \quad (1)$$

$$\hat{\pi}^{\mathrm{bal\text{-}DR}} \in \mathrm{argmin}_\pi \{ \hat{\tau}_{W, \hat{\mu}} + \lambda \mathfrak{E}(W, \pi; \|\cdot\|, \Lambda) : \pi \in \Pi, W \in \mathrm{argmin}_{W \in \mathcal{W}} \, \mathfrak{E}^2(W, \pi; \|\cdot\|, \Lambda) \} \quad (2)$$

The regularization term regularizes *both* the *balance* (*i.e.*, worst-case/posterior bias) that is achievable for $\pi$ *and* the *variance* in evaluating $\pi$. We include this regularizer for completeness and motivated by the results of [42] (which regularize variance), but find that it not necessary to include it in practice.

## 3.1 Optimizing the Balanced Policy Learner

Unlike [1, 7, 13, 41, 45], our (nonconvex) policy optimization problem does *not* reduce to weighted classification precisely because our weights are *not* multiplies of $\pi_{T_i}(X_i)$ (but therefore our weights also lead to better performance). Instead, like [42], we use gradient descent. For that, we need to be able to differentiate our bilevel optimization problem. We focus on $p = 2$ for brevity.

**Theorem 5.** *Let* $\| \cdot \| = \| \cdot \|_{2,\mathcal{K}_{1:m},\gamma_{1:m}}$. *Then* $\exists W^*(\pi) \in \operatorname{argmin}_{W \in \mathcal{W}} \mathfrak{E}^2(W, \pi; \| \cdot \|, \Lambda)$ *such that*

$$\nabla_{\pi_t(X_1),...,\pi_t(X_n)} \hat{\tau}_{W^*(\pi)} = \tfrac{1}{n} Y_{1:n}^T \tilde{H}(I - (A + (I-A)\tilde{H})^{-1}(I-A)\tilde{H})J_t$$

$$\nabla_{\pi_t(X_1),...,\pi_t(X_n)} \hat{\tau}_{W^*(\pi),\hat{\mu}} = \tfrac{1}{n} \hat{\epsilon}_{1:n}^T \tilde{H}(I - (A + (I-A)\tilde{H})^{-1}(I-A)\tilde{H})J_t + \tfrac{1}{n}\hat{\mu}_t(X_{1:n})$$

$$\nabla_{\pi_t(X_1),...,\pi_t(X_n)} \mathfrak{E}(W^*(\pi), \pi; \| \cdot \|, \Lambda) = -D_t / \mathfrak{E}(W^*(\pi), \pi; \| \cdot \|, \Lambda)$$

*where* $\tilde{H} = -F(F^T H F)^{-1} F^T$, $F_{ij} = \delta_{ij} - \delta_{in}$ *for* $i \in [n], j \in [n-1]$, $A_{ij} = \delta_{ij}\mathbb{I}[W_i^*(\pi) > 0]$, $D_{ti} = \gamma_t^2 \sum_{j=1}^n \mathcal{K}_t(X_i, X_j)(W_j \delta_{T_j t} - \pi_t(X_j))$, $H_{ij} = 2\sum_{t=1}^m \gamma_t^2 \delta_{T_i t}\delta_{T_j t}\mathcal{K}_t(X_i, X_j) + 2\Lambda$, *and* $J_{tij} = -2\gamma_t^2 \delta_{T_i t}\mathcal{K}_t(X_i, X_j)$.

To leverage this result, we use a parameterized policy class such as $\Pi_{\text{logit}} = \{\pi_t(x; \beta_t) \propto \exp(\beta_{t0} + \beta_t^T x)\}$ (or kernelized versions thereof), apply chain rule to differentiate objective in the parameters $\beta$, and use BFGS [15] with random starts. The logistic parametrization allows us to smooth the problem even while the solution ends up being deterministic (extreme $\beta$).

This approach requires solving a quadratic program for each objective gradient evaluation. While this can be made faster by using the previous solution as warm start, it is still computationally intensive, especially as the bilevel problem is nonconvex and both it and each quadratic program solved in "batch" mode. This is a limitation of the current optimization algorithm that we hope to improve on in the future using specialized methods for bilevel optimization [4, 32, 37].

**Example 2.** We return to Ex. 1 and consider policy learning. We use the fixed draw shown in Fig. 1a and set $\sigma$ to 0. We consider a variety of policy learners and plot the policies in Fig. 2 along with their population regret $\text{PAPE}(\pi) - \text{PAPE}(\pi^*)$. The policy learners we consider are: minimizing standard IPW and DR evaluations over $\Pi_{\text{logit}}$ with $\hat{\varphi}, \hat{\mu}$ as in Ex. 1 (versions with combinations of normalized, clipped, and/or true $\varphi$, not shown, all have regret 0.26–0.5), the direct method with Gaussian process regression gradient boosted trees (both sklearn defaults), weighted SVM classification using IPW and DR weights (details in supplement), SNPOEM [43], PF [23], and our balanced policy learner (1) with parameters as in Ex. 1, $\Pi = \Pi_{\text{logit}}$, $\lambda = \Lambda = 0$ (the DR version (2), not shown, has regret .08).

**Example 3.** Next, we consider two UCI multi-class classification datasets [30], Glass ($n = 214$, $d = 9$, $m = 6$) and Ecoli ($n = 336$, $d = 7$, $m = 8$), and use a supervised-to-contextual-bandit transformation [7, 13, 42] to compare different policy learning algorithms. Given a supervised multi-class dataset, we draw $T$ as per a multilogit model with random $\pm 1$ coefficients in the normalized covariates $X$. Further, we set $Y$ to 0 if $T$ matches the label and 1 otherwise. And we split the data 75-25 into training and test sample. Using 100 replications of this process, we evaluate the performance of learned *linear* policies by comparing the linear policy learners as in Ex. 2. For IPW-based approaches, we estimate $\hat{\varphi}$ by a multilogit regression (well-specified by construction). For DR approaches, we estimate $\hat{\mu}$ using gradient boosting trees (sklearn defaults). We compare these to our balanced policy learner in both vanilla and DR forms with all parameters fit by marginal likelihood using the RBF kernel with an unspecified length scale after normalizing the data. We tabulate the results in Tab. 2. They first demonstrate that employing the various stopgap fixes to IPW-based policy learning as in SNPOEM indeed provides a critical edge. This is further improved upon by using a balanced approach to policy learning, which gives the best results. In this example, DR approaches do worse than vanilla ones, suggesting both that XGBoost provided a bad outcome model and/or that the additional variance of DR was not compensated for by sufficiently less bias.

## 3.2 Uniform Consistency and Regret Bounds

Next, we establish consistency results *uniformly* over policy classes. This allows us to bound the regret of the balanced policy learner. We define the sample and population regret, respectively, as

$$R_\Pi(\hat{\pi}) = \text{PAPE}(\hat{\pi}) - \min_{\pi \in \Pi} \text{PAPE}(\pi), \quad \widehat{R}_\Pi(\hat{\pi}) = \text{SAPE}(\hat{\pi}) - \min_{\pi \in \Pi} \text{SAPE}(\pi)$$

Table 2: Policy learning results in Ex. 3

|       | IPW   | DR    | IPW-SVM | DR-SVM | POEM  | SNPOEM | Balanced | Balanced-DR |
|-------|-------|-------|---------|--------|-------|--------|----------|-------------|
| Glass | 0.726 | 0.755 | 0.641   | 0.731  | 0.851 | 0.615  | **0.584**| 0.660       |
| Ecoli | 0.488 | 0.501 | 0.332   | 0.509  | 0.431 | 0.331  | **0.298**| 0.371       |

A key requirement for these to converge is that the best-in-class policy is learnable. We quantify that using Rademacher complexity [3] and later extend our results to VC dimension. Let us define

$$\widehat{\mathfrak{R}}_n(\mathcal{F}) = \tfrac{1}{2^n} \sum_{\rho_i \in \{-1,+1\}^n} \sup_{f \in \mathcal{F}} \tfrac{1}{n} \sum_{i=1}^n \rho_i f(X_i), \quad \mathfrak{R}_n(\mathcal{F}) = \mathbb{E}[\widehat{\mathfrak{R}}_n(\mathcal{F})].$$

*E.g.*, for linear policies $\widehat{\mathfrak{R}}_n(\mathcal{F}) = O(1/\sqrt{n})$ [21]. If $\mathcal{F} \subseteq [\mathcal{X} \to \mathbb{R}^m]$ let $\mathcal{F}_t = \{(f(\cdot))_t : f \in \mathcal{F}\}$ and set $\mathfrak{R}_n(\mathcal{F}) = \sum_{t=1}^m \mathfrak{R}_n(\mathcal{F}_t)$ and same for $\widehat{\mathfrak{R}}_n(\mathcal{F})$. We also strengthen the overlap assumption.

**Assumption 3** (Strong overlap). $\exists \alpha \geq 1$ such that $\mathbb{P}(\varphi_t(X) \geq 1/\alpha) = 1 \ \forall t \in [m]$.

**Theorem 6.** *Fix $\Pi \subseteq [\mathcal{X} \to \Delta^m]$ and let $W_n^*(\pi) = W_n^*(\pi; \|f\|_{p, \mathcal{K}_{1:m}, \gamma_{n,1:m}}, \Lambda_n)$ with $0 \prec \underline{\kappa} I \preceq \Lambda_n \preceq \overline{\kappa} I$, $0 < \underline{\gamma} \leq \gamma_{n,t} \leq \overline{\gamma} \ \forall t \in [m]$ for each $n$ and $\pi \in \Pi$. Suppose Asns. 1 and 3 hold, $|\epsilon_i| \leq B$ a.s. bounded, and $\sqrt{\mathcal{K}_t(x,x)} \leq \Gamma \ \forall t \in [m]$ for $\Gamma \geq 1$. Then the following two results hold:*

(a) *If $\|\mu_t\|_{\mathcal{K}_t} < \infty$, $\forall t \in [m]$ then for $n$ sufficiently large ($n \geq 2\log(4m/\nu)/(1/(2\alpha) - \mathfrak{R}_n(\Pi))^2$), we have that, with probability at least $1 - \nu$,*

$$
\begin{aligned}
\sup_{\pi \in \Pi} |\hat{\tau}_{W^*(\pi)} - \text{SAPE}(\pi)| \leq\ &8\alpha\Gamma\overline{\gamma}m(\|\mu\| + \sqrt{2\log(4m/\nu)}\underline{\kappa}^{-1}B)\mathfrak{R}_n(\Pi) \\
&+ \tfrac{1}{\sqrt{n}}\left(2\alpha\overline{\kappa}\|\mu\| + 12\alpha\Gamma^2\overline{\gamma}m\|\mu\| + 6\alpha\Gamma\overline{\gamma}m\underline{\kappa}^{-1}B\log\left(\tfrac{4m}{\nu}\right)\right) \\
&+ \tfrac{1}{\sqrt{n}}(2\alpha\overline{\kappa}\underline{\kappa}^{-1}B + 12\alpha\Gamma^2\overline{\gamma}m\underline{\kappa}^{-1}B + 3\alpha\Gamma\overline{\gamma}m\|\mu\|)\sqrt{2\log\left(\tfrac{4m}{\nu}\right)}
\end{aligned}
$$

(b) *If $\mathcal{K}_t$ is $C_0$-universal for all $t \in [m]$ and either $\mathfrak{R}_n(\Pi) = o(1)$ or $\widehat{\mathfrak{R}}_n(\Pi) = o_p(1)$ then*

$$\sup_{\pi \in \Pi} |\hat{\tau}_{W^*(\pi)} - \text{SAPE}(\pi)| = o_p(1).$$

The proof crucially depends on *simultaneously* handling the functional complexities of both the policy class $\Pi$ *and* the space of functions $\{f : \|f\| < \infty\}$ being balanced against. Again, the key assumptions of Thm. 6 are unconfoundedness, overlap, and bounded residuals. The other conditions simply guide the choice of method parameters. Regret bounds follow as a corollary.

**Corollary 7.** *Suppose the assumptions of Thm. 6 hold. If $\hat{\pi}_n^{bal}$ is as in (1) then:*

(a) *If $\|\mu_t\|_{\mathcal{K}_t} < \infty$ for all $t \in [m]$:* $\qquad\qquad R_\Pi(\hat{\pi}_n^{bal}) = O_p(\mathfrak{R}_n(\Pi) + 1/\sqrt{n})$.

(b) *If $\mathcal{K}_t$ is $C_0$-universal for all $t \in [m]$:* $\qquad\qquad R_\Pi(\hat{\pi}_n^{bal}) = o_p(1)$.

*If $\hat{\pi}_n^{bal\text{-}DR}$ is as in (2) then:*

(c) *If $\|\hat{\mu}_{nt} - \mu_t\|_{\mathcal{K}_t} = o_p(1)$ for all $t \in [m]$:* $\qquad R_\Pi(\hat{\pi}_n^{bal\text{-}DR}) = O_p(\mathfrak{R}_n(\Pi) + 1/\sqrt{n})$.

(d) *If $\|\hat{\mu}_n(X) - \mu(X)\|_2 = O_p(r(n))$:* $\qquad\qquad R_\Pi(\hat{\pi}_n^{bal\text{-}DR}) = O_p(r(n) + \mathfrak{R}_n(\Pi) + 1/\sqrt{n})$.

(e) *If $\|\mu_t\|_{\mathcal{K}_t} < \infty, \|\hat{\mu}_{nt}\|_{\mathcal{K}_t} = O_p(1)$ for all $t \in [m]$:* $\quad R_\Pi(\hat{\pi}_n^{bal\text{-}DR}) = O_p(\mathfrak{R}_n(\Pi) + 1/\sqrt{n})$.

(f) *If $\mathcal{K}_t$ is $C_0$-universal for all $t \in [m]$:* $\qquad\qquad R_\Pi(\hat{\pi}_n^{bal}) = o_p(1)$.

*And, all the same results hold when replacing $\mathfrak{R}_n(\Pi)$ with $\widehat{\mathfrak{R}}_n(\Pi)$ and/or replacing $R_\Pi$ with $\widehat{R}_\Pi$.*

## 4 Conclusion

Considering the policy evaluation and learning problems using observational or logged data, we presented a new method that is based on finding optimal balancing weights that make the data look like the target policy and that is aimed at ameliorating the shortcomings of existing methods, which included having to deal with near-zero propensities, using too few positive weights, and using an awkward two-stage procedure. The new approach showed promising signs of fixing these issues in some numerical examples. However, the new learning method is more computationally intensive than existing approaches, solving a QP at each gradient step. Therefore, in future work, we plan to explore faster algorithms that can implement the balanced policy learner, perhaps using alternating descent, and use these to investigate comparative numerics in much larger datasets.

## Acknowledgements

This material is based upon work supported by the National Science Foundation under Grant No. 1656996.

## Footnotes

[1]Thus, although the data is iid, the $t$-treated sample $\{i : T_i = t\}$ may differ *systematically* from the $t'$-treated sample $\{i : T_i = t'\}$ for $t \neq t'$, *i.e.*, not necessarily just by chance as in a randomized controlled trial (RCT).

[2] This problem is *unique* to policy evaluation and learning – in causal effect estimation, the IPW estimator for SATE has nonzero weights on *all* of the data points. For policy learning with $m = 2$, Athey and Wager [1], Beygelzimer and Langford [7] minimize estimates of the form $\frac{1}{2}(\hat{\tau}(\pi) - \hat{\tau}(1(\cdot) - \pi))$ with $\hat{\tau}(\pi) = \hat{\tau}^{\text{IPW}}(\pi)$ or $= \hat{\tau}^{\text{DR}}(\pi)$. This evaluates $\pi$ *relative* to the uniformly random policy and the resulting total weighted sums over $Y_i$ or $\hat{\epsilon}_i$ have nonzero weights whether $\pi_{T_i}(X_i) = 0$ or not. While a useful approach for reduction to weighted classification [7] or invoking semi-parametric theory [1], it only works for $m = 2$, has no effect on learning as the centering correction is constant in $\pi$, and, for evaluation, is not an estimator for SAPE.

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
