[Supplementary Material]

## A  Omitted Proofs

*Proof of Thm. 1.* Noting that $Y_i = Y_i(T_i) = \sum_{t=1}^m \delta_{T_i t}\mu_t(X_i) + \epsilon_i$, let us rewrite $\hat{\tau}_W$ as

$$\hat{\tau}_W = \tfrac{1}{n}\sum_{t=1}^m \sum_{i=1}^n W_i \delta_{T_i t}\mu_t(X_i) + \tfrac{1}{n}\sum_{i=1}^n W_i \epsilon_i.$$

Recalling that $\mathrm{SAPE}(\pi) = \tfrac{1}{n}\sum_{i=1}^n \sum_{t=1}^m \pi_t(X_i)\mu_t(X_i)$ immediately yields the first result. To obtain the second result note that $\mathrm{SAPE}(\pi)$ is measurable with respect to $X_{1:n}, T_{1:n}$ so that

$$\mathrm{CMSE}(\hat{\tau}_W, \pi) = (\mathbb{E}[\hat{\tau}_W \mid X_{1:n}, T_{1:n}] - \mathrm{SAPE}(\pi))^2 + \mathrm{Var}(\hat{\tau}_W \mid X_{1:n}, T_{1:n}).$$

By Asn. 1

$$\mathbb{E}[\delta_{T_i t}\epsilon_i \mid X_{1:n}, T_{1:n}] = \delta_{T_i t}\mathbb{E}[\epsilon_i \mid X_i] = \delta_{T_i t}(\mathbb{E}[Y_i(t) \mid X_i] - \mu_t(X_i)) = 0.$$

Therefore,

$$\mathbb{E}[\hat{\tau}_W \mid X_{1:n}, T_{1:n}] = \tfrac{1}{n}\sum_{t=1}^m \sum_{i=1}^n W_i \delta_{T_i t}\mu_t(X_i),$$

giving the first term of $\mathrm{CMSE}(\hat{\tau}_W, \pi)$. Moreover, since

$$\mathbb{E}[\epsilon_i \epsilon_{i'} \mid X_{1:n}, T_{1:n}] = \delta_{ii'}\sigma_{T_i}^2,$$

we have

$$\mathrm{Var}(\hat{\tau}_W \mid X_{1:n}, T_{1:n}) = \mathbb{E}[(\hat{\tau}_W - \mathbb{E}[\hat{\tau}_W \mid X_{1:n}, T_{1:n}])^2 \mid X_{1:n}, T_{1:n}]$$
$$= \tfrac{1}{n^2}\mathbb{E}[(\sum_{i=1}^n W_i \epsilon_i)^2 \mid X_{1:n}, T_{1:n}] = \tfrac{1}{n^2}\sum_{i=1}^n W_i^2 \sigma_{T_i}^2,$$

giving the second term. $\qquad\square$

*Proof of Cor. 2.* This follows from Thm. 1 after noting that $\hat{\tau}_{W,\hat{\mu}} = \hat{\tau}_W - B(W, \pi; \hat{\mu})$ and that $B_t(W, \pi; \hat{\mu}_t) - B_t(W, \pi; \hat{\mu}_t) = B_t(W, \pi; \mu_t - \hat{\mu}_t)$. $\qquad\square$

*Proof of Lemma 1.* For the first statement, we have

$$\mathfrak{E}^2(W, \pi; \|\cdot\|_{p,\mathcal{K}_{1:m},\gamma_{1:m}}, \Lambda) = \sup_{\|v\|_p \leq 1, \|f_t\|_{\mathcal{K}_t} \leq \gamma_t v_t}(\sum_{t=1}^m B_t(W, \pi; f_t))^2 + \tfrac{1}{n^2}W^T \Lambda W$$
$$= \sup_{\|v\|_p \leq 1}(\sum_{t=1}^m \sup_{\|f_t\|_{\mathcal{K}_t} \leq \gamma_t v_t} B_t(W, \pi; f_t))^2 + \tfrac{1}{n^2}W^T \Lambda W$$
$$= \sup_{\|v\|_p \leq 1}(\sum_{t=1}^m v_t \gamma_t \mathfrak{B}_t(W, \pi; \|\cdot\|_{\mathcal{K}_t}))^2 + \tfrac{1}{n^2}W^T \Lambda W$$
$$= (\sum_{t=1}^m \gamma_t^q \mathfrak{B}_t^q(W, \pi; \|\cdot\|_{\mathcal{K}_t}))^{2/q} + \tfrac{1}{n^2}W^T \Lambda W.$$

For the second statement, let $z_{ti} = (W_i \delta_{T_i t} - \pi_t(X_i))$ and note that since $\mathbb{E}[(\mu_t(X_i) - f_t(X_i))(\mu_s(X_j) - f_s(X_j)) \mid X_{1:n}, T_{1:n}] = \delta_{ts}\mathcal{K}_t(X_i, X_j)$, we have

$$\mathrm{CMSE}(\hat{\tau}_{W,f}, \pi) = \mathbb{E}[(\sum_{t=1}^m B_t(W, \pi_t; \mu_t - f_t))^2 \mid X_{1:n}, T_{1:n}] + \tfrac{1}{n^2}W^T \Sigma W$$
$$= \sum_{t,s=1}^m \sum_{i,j=1}^n z_{ti}z_{sj}\mathbb{E}[(\mu_t(X_i) - f_t(X_i))(\mu_s(X_j) - f_s(X_j)) \mid X_{1:n}, T_{1:n}]$$
$$+ \tfrac{1}{n^2}W^T \Sigma W$$
$$= \sum_{t=1}^m \sum_{i,j=1}^n z_{ti}z_{tj}\mathcal{K}_t(X_i, X_j) + \tfrac{1}{n^2}W^T \Sigma W.$$

$\qquad\square$

*Proof of Thm. 3.* Let $Z = \tfrac{1}{n}\sum_{i=1}^n \pi_{T_i}(X_i)/\varphi_{T_i}(X_i)$ and $\tilde{W}_i(\pi) = \tfrac{1}{Z}\pi_{T_i}(X_i)/\varphi_{T_i}(X_i)$ and note that $\tilde{W} \in \mathcal{W}$. Moreover, note that

$$\mathfrak{B}_t(\tilde{W}, \pi_t; \|\cdot\|_{\mathcal{K}_t}) = \tfrac{1}{Z}\|\tfrac{1}{n}\sum_{i=1}^n(\tfrac{\delta_{T_i t}}{\varphi_t(X_i)} - Z)\pi_t(X_i)E_{X_i}\|_{\mathcal{K}_t}$$
$$\leq \tfrac{1}{Z}\|\tfrac{1}{n}\sum_{i=1}^n(\tfrac{\delta_{T_i t}}{\varphi_t(X_i)} - 1)\pi_t(X_i)E_{X_i}\|_{\mathcal{K}_t} + \tfrac{1}{Z}\|\tfrac{1}{n}\sum_{i=1}^n(Z-1)\pi_t(X_i)E_{X_i}\|_{\mathcal{K}_t}$$
$$\leq \tfrac{1}{Z}\|\tfrac{1}{n}\sum_{i=1}^n(\tfrac{\delta_{T_i t}}{\varphi_t(X_i)} - 1)\pi_t(X_i)E_{X_i}\|_{\mathcal{K}_t} + \tfrac{|Z-1|}{Z}\tfrac{1}{n}\sum_{i=1}^n \sqrt{\mathcal{K}_t(X_i, X_i)}.$$

Let $\xi_i = (\tfrac{\delta_{T_i t}}{\varphi_t(X_i)} - 1)\pi_t(X_i)E_{X_i}$ and note that $\mathbb{E}[\xi_i] = \mathbb{E}[(\mathbb{E}[\delta_{T_i t}/\varphi_t(X_i) \mid X_i] - 1)\pi_t(X_i)E_{X_i}] = 0$ and that $\xi_1, \xi_2, \dots$ are iid. Therefore, letting $\xi_1', \xi_2', \dots$ be iid replicates of $\xi_1, \xi_2, \dots$ (ghost sample) and letting $\rho_i$ be iid Rademacher random variables independent of all else, we have

$$\mathbb{E}[\|\tfrac{1}{n}\sum_{i=1}^n \xi_i\|_{\mathcal{K}_t}^2] = \tfrac{1}{n^2}\mathbb{E}[\|\sum_{i=1}^n(\mathbb{E}[\xi_i'] - \xi_i)\|_{\mathcal{K}_t}^2] \leq \tfrac{1}{n^2}\mathbb{E}[\|\sum_{i=1}^n(\xi_i' - \xi_i)\|_{\mathcal{K}_t}^2]$$
$$= \tfrac{1}{n^2}\mathbb{E}[\|\sum_{i=1}^n \rho_i(\xi_i' - \xi_i)\|_{\mathcal{K}_t}^2] \leq \tfrac{4}{n^2}\mathbb{E}[\|\sum_{i=1}^n \rho_i\xi_i\|_{\mathcal{K}_t}^2]$$

Note that $\|\xi_1 - \xi_2\|_{\mathcal{K}_t}^2 + \|\xi_1 + \xi_2\|_{\mathcal{K}_t}^2 = 2\|\xi_1\|_{\mathcal{K}_t}^2 + 2\|\xi_2\|_{\mathcal{K}_t}^2 + 2\langle \xi_1, \xi_2\rangle - 2\langle \xi_1, \xi_2\rangle = 2\|\xi_1\|_{\mathcal{K}_t}^2 + 2\|\xi_2\|_{\mathcal{K}_t}^2$. By induction, $\sum_{\rho_i \in \{-1,+1\}^n} \|\sum_{i=1}^n \rho_i \xi_i\|_{\mathcal{K}_t}^2 = 2^n \sum_{i=1}^n \|\xi_i\|_{\mathcal{K}_t}^2$. Since

$$\mathbb{E}[\|\xi_i\|_{\mathcal{K}_t}^2] \leq 2\mathbb{E}[\tfrac{\pi_T^2(X)}{\varphi_T^2(X)}\mathcal{K}_t(X,X)] + 2\mathbb{E}[\pi_t^2(X)\mathcal{K}_t(X,X)] \leq 4\mathbb{E}[\tfrac{\pi_T^2(X)}{\varphi_T^2(X)}\mathcal{K}_t(X,X)] < \infty,$$

we get $\mathbb{E}[\|\frac{1}{n}\sum_{i=1}^n \xi_i\|_{\mathcal{K}_t}^2] = O(1/n)$ and therefore $\|\frac{1}{n}\sum_{i=1}^n \xi_i\|_{\mathcal{K}_t}^2 = O_p(1/n)$ by Markov's inequality. Moreover, as $\mathbb{E}[\pi_T(X)/\varphi_T(X)] = \mathbb{E}[\sum_{t=1}^m \mathbb{E}[\delta_{Tt} \mid X]\pi_t(X)/\varphi_t(X)] = \mathbb{E}[\sum_{t=1}^m \pi_t(X)] = 1$ and $\mathbb{E}[\pi_T^2(X)/\varphi_T(X)^2] < \infty$, by Chebyshev's inequality, $\mathbb{E}[(Z-1)^2] = O(1/n)$ so that $(Z-1)^2 = O_p(1/n)$ by Markov's inequality. Similarly, as $\mathbb{E}[\sqrt{\mathcal{K}_t(X,X)}] < \infty$, we have $\frac{1}{n}\sum_{i=1}^n \sqrt{\mathcal{K}_t(X_i,X_i)} \to_p \mathbb{E}[\sqrt{\mathcal{K}_t(X,X)}]$. Putting it all together, by Slutsky's theorem, $\mathfrak{B}_t^2(\tilde{W}, \pi_t; \|\cdot\|_{\mathcal{K}_t}) = O_p(1/n)$. Moreover, $\|\tilde{W}\|_2^2 = \frac{1}{Z^2}\sum_{i=1}^n \pi_{T_i}^2(X_i)/\varphi_{T_i}^2(X_i) = O_p(n)$. Therefore, since $\Lambda_n \preceq \overline{\kappa}I$ and since $W_n^*$ is optimal and $\tilde{W} \in \mathcal{W}$, we have

$$\mathfrak{E}^2(W_n^*, \pi; \|\cdot\|_{p,\mathcal{K}_{1:m},\gamma_{n,1:m}}, \Lambda_n) \leq \mathfrak{E}^2(\tilde{W}, \pi; \|\cdot\|_{p,\mathcal{K}_{1:m},\gamma_{n,1:m}}, \Lambda_n)$$

$$\leq \overline{\gamma}^2\left(\sum_{t=1}^m \mathfrak{B}_t^q(\tilde{W}, \pi_t; \|\cdot\|_{\mathcal{K}_t})\right)^{2/q} + \tfrac{\overline{\kappa}}{n^2}\|\tilde{W}\|_2^2 = O_p(1/n)$$

Therefore,

$$\mathfrak{B}_t^2(W_n^*, \pi_t; \|\cdot\|_{\mathcal{K}_t}) \leq \underline{\gamma}^{-2}\mathfrak{E}^2(W_n^*, \pi; \|\cdot\|_{1:m}, \gamma_{n,1:m}, \Lambda_n) = O_p(1/n),$$

$$\tfrac{1}{n^2}\|W_n^*\|_2^2 \leq \tfrac{1}{\underline{\kappa}n^2}W_n^{*T}\Lambda_n W_n^* \leq \underline{\kappa}^{-1}\mathfrak{E}^2(W_n^*, \pi; \|\cdot\|_{1:m}, \gamma_{n,1:m}, \Lambda_n) = O_p(1/n).$$

Now consider case (a). By assumption $\|\Sigma\|_2 \leq \overline{\sigma}^2 < \infty$ for all $n$. Then we have

$$\text{CMSE}(\hat{\tau}_{W_n^*}, \pi) \leq m\sum_{t=1}^m \|\mu_t\|_{\mathcal{K}_t}^2 \mathfrak{B}_t^2(W_n^*, \pi_t; \|\cdot\|_{\mathcal{K}_t}) + \tfrac{\overline{\sigma}^2}{n^2}\|W_n^*\|_2^2 = O_p(1/n).$$

Letting $D_n = \sqrt{n}\left|\hat{\tau}_{W_n^*} - \text{SAPE}(\pi)\right|$ and $\mathcal{G}$ be the sigma algebra of $X_1, T_1, X_2, T_2, \ldots$, Jensen's inequality yields $\mathbb{E}[D_n \mid \mathcal{G}] = O_p(1)$ from the above. We proceed to show that $D_n = O_p(1)$, yielding the first result. Let $\nu > 0$ be given. Then $\mathbb{E}[D_n \mid \mathcal{G}] = O_p(1)$ says that there exist $N, M$ such that $\mathbb{P}(\mathbb{E}[D_n \mid \mathcal{G}] > M) \leq \nu/2$ for all $n \geq N$. Let $M_0 = \max\{M, 2/\nu\}$ and observe that, for all $n \geq N$,

$$\mathbb{P}(D_n > M_0^2) = \mathbb{P}(D_n > M_0^2, \mathbb{E}[D_n \mid \mathcal{G}] > M_0) + \mathbb{P}(D_n > M_0^2, \mathbb{E}[D_n \mid \mathcal{G}] \leq M_0)$$

$$= \mathbb{P}(D_n > M_0^2, \mathbb{E}[D_n \mid \mathcal{G}] > M_0) + \mathbb{E}[\mathbb{P}(D_n > M_0^2 \mid \mathcal{G})\mathbb{I}[\mathbb{E}[D_n \mid \mathcal{G}] \leq M_0]]$$

$$\leq \nu/2 + \mathbb{E}[\tfrac{\mathbb{E}[D_n|\mathcal{G}]}{M_0^2}\mathbb{I}[\mathbb{E}[D_n \mid \mathcal{G}] \leq M_0]] \leq \nu/2 + 1/M_0 \leq \nu$$

Now consider case (b). We first show that $B_t(W_n^*, \pi_t; \mu_t) = o_p(1)$. Fix $t \in [m]$ and $\eta > 0, \nu > 0$. Because $\mathfrak{B}_t(W_n^*, \pi_t; \|\cdot\|_{\mathcal{K}_t}) = O_p(n^{-1/2}) = o_p(n^{-1/4})$ and $\|W_n^*\|_2 = O_p(\sqrt{n})$, there are $M, N$ such that for all $n \geq N$ both $\mathbb{P}(n^{1/4}\mathfrak{B}_t(W_n^*, \pi_t; \|\cdot\|_{\mathcal{K}_t}) > \sqrt{\eta}) \leq \nu/3$ and $\mathbb{P}(n^{-1/2}\|W_n^*\|_2 > M\sqrt{\eta}) \leq \nu/3$. Next, fix $\tau = \sqrt{\nu\eta/3}/M$. By existence of second moment, there is $g_0' = \sum_{i=1}^\ell \beta_i I_{S_i}$ with $(\mathbb{E}[(\mu_t(X) - g_0'(X))^2])^{1/2} \leq \tau/2$ where $I_S(x)$ are the simple functions $I_S(x) = \mathbb{I}[x \in S]$ for $S$ measurable. Let $i = 1, \ldots, \ell$. Let $U_i \supset S_i$ open and $E_i \subseteq S_i$ compact be such that $\mathbb{P}(X \in U_i \backslash E_i) \leq \tau^2/(4\ell|\beta_i|)^2$. By Urysohn's lemma [36], there exists a continuous function $h_i$ with support $C_i \subseteq U_i$ compact, $0 \leq h_i \leq 1$, and $h_i(x) = 1 \forall x \in E_i$. Therefore, $(\mathbb{E}[(I_{S_i}(X) - h_i)^2])^{1/2} = (\mathbb{E}[(I_{S_i}(X) - h_i)^2\mathbb{I}[X \in U_i \backslash E_i]])^{1/2} \leq (\mathbb{P}(X \in U_i \backslash E_i))^{1/2} \leq \tau/(4\ell|\beta_i|)$. By $C_0$-universality, $\exists g_i = \sum_{j=1}^m \alpha_j \mathcal{K}_t(x_j, \cdot)$ such that $\sup_{x \in \mathcal{X}}|h_i(x) - g_i(x)| < \tau/(4\ell|\beta_i|)$. Because $\mathbb{E}[(h_i - g_i)^2] \leq \sup_{x \in \mathcal{X}}|h_i(x) - g_i(x)|^2$, we have $\sqrt{\mathbb{E}[(I_{S'}(X) - g_i)^2]} \leq \tau/(2\ell|\beta_i|)$. Let $\tilde{\mu}_t = \sum_{i=1}^\ell \beta_i g_i$. Then $(\mathbb{E}[(\mu_t(X) - \tilde{\mu}_t(X))^2])^{1/2} \leq \tau/2 + \sum_{i=1}^\ell |\beta_i|\tau/(2\ell|\beta_i|) = \tau$ and $\|\tilde{\mu}_t\|_{\mathcal{K}_t} < \infty$. Let $\delta_n = \sqrt{\frac{1}{n}\sum_{i=1}^n(\mu_t(X_i) - \tilde{\mu}_t(X_i))^2}$ so that $\mathbb{E}\delta_n^2 \leq \tau^2$. Now, because we have

$$B_t(W_n^*, \pi_t; \mu_t) = B_t(W_n^*, \pi_t; \tilde{\mu}_t) + B_t(W_n^*, \pi_t; \mu_t - \tilde{\mu}_t)$$

$$\leq \|\tilde{\mu}_t\|_{\mathcal{K}_t}\mathfrak{B}_t(W_n^*, \pi_t; \|\cdot\|_{\mathcal{K}_t}) + \sqrt{\tfrac{1}{n}\sum_{i=1}^n(W_{ni}^*\delta_{T_it} - \pi_t(X_i))^2}\delta_n$$

$$\leq \|\tilde{\mu}_t\|_{\mathcal{K}_t}\mathfrak{B}_t(W_n^*, \pi_t; \|\cdot\|_{\mathcal{K}_t}) + (n^{-1/2}\|W_n^*\|_2 + 1)\delta_n,$$

letting $N' = \max\{N, 2\lceil\|\tilde{\mu}_t\|_{\mathcal{K}_t}^4/\eta^2\rceil\}$, we must then have, for all $n \geq N'$, by union bound and by Markov's inequality, that

$$\mathbb{P}(B_t(W_n^*, \pi_t; \mu_t) > \eta) \leq \mathbb{P}(n^{-1/4}\|\tilde{\mu}_t\|_{\mathcal{K}_t} > \sqrt{\eta}) + \mathbb{P}(n^{1/4}\mathfrak{B}_t(W_n^*, \pi_t; \|\cdot\|_{\mathcal{K}_t}) > \sqrt{\eta})$$
$$+ \mathbb{P}(n^{-1/2}\|W_n^*\|_2 > M\sqrt{\eta}) \leq \nu/3 + \mathbb{P}(\delta_n > \sqrt{\eta}/M)$$
$$\leq 0 + \nu/3 + \nu/3 + \nu/3 = \nu.$$

Following the same logic as in case (a), we get $\text{CMSE}(\hat{\tau}_{W_n^*}, \pi) = o_p(1)$, so letting $D_n = |\hat{\tau}_{W_n^*} - \text{SAPE}(\pi)|$ and $\mathcal{G}$ be as before, we have $\mathbb{E}[D_n \mid \mathcal{G}] = o_p(1)$ by Jensen's inequality. Let $\eta > 0, \nu > 0$ be given. Let $N$ be such that $\mathbb{P}(\mathbb{E}[D_n \mid \mathcal{G}] > \nu\eta/2) \leq \nu/2$. Then for all $n \geq N$:

$$\mathbb{P}(D_n > \eta) = \mathbb{P}(D_n > \eta, \mathbb{E}[D_n \mid \mathcal{G}] > \eta\nu/2) + \mathbb{P}(D_n > \eta, \mathbb{E}[D_n \mid \mathcal{G}] \leq \eta\nu/2)$$
$$= \mathbb{P}(D_n > \eta, \mathbb{E}[D_n \mid \mathcal{G}] > \eta\nu/2) + \mathbb{E}[\mathbb{P}(D_n > \eta \mid \mathcal{G})\mathbb{I}[\mathbb{E}[D_n \mid \mathcal{G}] \leq \eta\nu/2]]$$
$$\leq \nu/2 + \mathbb{E}[\tfrac{\mathbb{E}[D_n|\mathcal{G}]}{\eta}\mathbb{I}[\mathbb{E}[D_n \mid \mathcal{G}] \leq \eta\nu/2]] \leq \nu/2 + \nu/2 \leq \nu,$$

showing that $D_n = o_p(1)$ and completing the proof. $\qquad\square$

*Proof of Cor. 4.* Case (a) follows directly from the proof of Thm. 3 noting that the bias term now disappears at rate $o_p(1)O_p(1/\sqrt{n}) = o_p(1/\sqrt{n})$. For Case (b), observe that by Cauchy-Schwartz and Slutsky's theorem $|B_t(W, \pi_t; \mu - \hat{\mu}_n)| \leq (n^{-1/2}\|W_n^*\|_2 + 1)(\frac{1}{n}\sum_{i=1}^n(\hat{\mu}_n(X_i) - \mu(X_i))^2)^{1/2} = O_p(r_n)$. For cases in cases (c) and (d) we treat $B_t(W, \pi_t; \mu - \hat{\mu}_n)$ as in the proof of Thm. 3 noting that $\|\mu_t - \hat{\mu}_{nt}\|_{\mathcal{K}_t} \leq \|\mu_t\|_{\mathcal{K}_t} + \|\hat{\mu}_{nt}\|_{\mathcal{K}_t}$ and that, in case (c), $\|\hat{\mu}_{nt}\|_{\mathcal{K}_t} = O_p(1)$ implies by Markov's inequality that $\|\hat{\mu}_{nt}\|_{\mathcal{K}_t} = O_p(1)$. The rest follows as in the proof of Thm. 3. $\qquad\square$

*Proof of Thm. 5.* First note that because our problem is a quadratic program, the KKT conditions are necessary and sufficient and we can always choose an optimizer where strict complementary slackness holds.

Ignore previous definitions of some symbols, consider any linearly constrained parametric nonlinear optimization problem in standard form: $z(x) \in \text{argmin}_{y \geq 0, By = b} f(x, y)$ where $x \in \mathbb{R}^n$, $y \in \mathbb{R}^m$, and $b \in \mathbb{R}^\ell$. KKT says there exist $\mu(x) \in \mathbb{R}^m, \lambda(x) \in \mathbb{R}^\ell$ such that (a) $\nabla_y f(x, z(x)) = \mu(x) + B^T\lambda(x)$, (b) $Bz(x) = b$, (c) $z(x) \geq 0$, (d) $\mu(x) \geq 0$, and (e) $\mu(x) \odot z(x) = 0$, where $\odot$ is the Hadamard product. Suppose strict complementary slackness holds in that (f) $\mu(x) + z(x) > 0$. By (a), we have that

$$\nabla_{xy}f(x, z(x)) + \nabla_{yy}f(x, z(x))\nabla_x z(x) = \nabla_x\mu(x) + B^T\nabla_x\lambda(x),$$

and hence, letting $H = \nabla_{yy}f(x, z(x))$ and $J = \nabla_{xy}f(x, z(x))$,

$$\nabla z(x) = H^{-1}(\nabla_x\mu(x) + B^T\nabla\lambda(x) - J).$$

By (b), we have that $B\nabla z(x) = 0$ so that

$$BH^{-1}\nabla_x\mu(x) + BH^{-1}B^T\nabla\lambda = BH^{-1}J,$$

and hence if the columns of $F$ form a basis for the null space of $B$ and $\tilde{H} = -F(F^THF)^{-1}F^T$,

$$\nabla_x z(x) = (H^{-1}B^T(AH^{-1}A^T)^{-1}AH^{-1} - H^{-1})(J - \nabla_x\mu(x)) = \tilde{H}(J - \nabla_x\mu(x)).$$

By (e), we have that

$$z_i(x)\nabla_x\mu_i(x) + \mu_i(x)\nabla_x z_i(x) = 0,$$

and then by (f), letting $A = \text{diag}(\mathbb{I}[z_1(x) > 0, \ldots, z_m(x) > 0])$ we have

$$A\nabla_x\mu(x) = 0, \quad (I - A)\nabla_x z(x) = 0,$$

and therefore

$$A\nabla_x\mu(x) - (I - A)\tilde{H}(J - \nabla_x\mu(x)) = 0$$

yielding finally that

$$\nabla_x z(x) = \tilde{H}(I - (A + (I - A)\tilde{H})^{-1}(I - A)\tilde{H})J.$$

The rest of the theorem is then begotten by applying this result and using chain rule. $\qquad\square$

*Proof of Thm. 6.* Let $Z(\pi) = \frac{1}{n}\sum_{i=1}^n \pi_{T_i}(X_i)/\varphi_{T_i}(X_i)$ and $\tilde{W}_i(\pi) = \frac{1}{Z(\pi)}\pi_{T_i}(X_i)/\varphi_{T_i}(X_i)$ and note that $\tilde{W} \in \mathcal{W}$. Moreover, note that

$$\sup_{\pi \in \Pi} \mathfrak{B}_t(\tilde{W}, \pi_t; \|\cdot\|_{\mathcal{K}_t}) = \sup_{\pi \in \Pi, \|f_t\|_{\mathcal{K}_t} \leq 1} \frac{1}{Z(\pi)} \frac{1}{n} \sum_{i=1}^n \left(\frac{\delta_{T_i t}}{\varphi_t(X_i)} - Z(\pi)\right)\pi_t(X_i)f_t(X_i)$$

$$\leq (\sup_{\pi \in \Pi} Z(\pi)^{-1}) \sup_{\pi_t \in \Pi_t, \|f_t\|_{\mathcal{K}_t} \leq 1} \frac{1}{n} \sum_{i=1}^n \left(\frac{\delta_{T_i t}}{\varphi_t(X_i)} - 1\right)\pi_t(X_i)f_t(X_i) + \Gamma \sup_{\pi \in \Pi} \left|1 - Z(\pi)^{-1}\right|.$$

We first treat the random variable

$$\Xi_t(X_{1:n}, T_{1:n}) = \sup_{\pi_t \in \Pi_t, \|f_t\|_{\mathcal{K}_t} \leq 1} \frac{1}{n} \sum_{i=1}^n \left(\frac{\delta_{T_i t}}{\varphi_t(X_i)} - 1\right)\pi_t(X_i)f_t(X_i).$$

Fix $x_{1:n}, t_{1:n}, x'_{1:n}, t'_{1:n}$ such that $x'_i = x_i, t'_i = t_i \, \forall i \neq i'$ and note that

$$\Xi_t(x_{1:n}, t_{1:n}) - \Xi_t(x'_{1:n}, t'_{1:n}) \leq \sup_{\pi_t \in \Pi_t, \|f_t\|_{\mathcal{K}_t} \leq 1} \left(\frac{1}{n}\sum_{i=1}^n \left(\frac{\delta_{t_i t}}{\varphi_t(x_i)} - 1\right)\pi_t(x_i)f_t(x_i)\right.$$

$$\left. - \frac{1}{n}\sum_{i=1}^n \left(\frac{\delta_{t'_i t}}{\varphi_t(x'_i)} - 1\right)\pi_t(x'_i)f_t(x'_i)\right)$$

$$= \frac{1}{n}\sup_{\pi_t \in \Pi_t, \|f_t\|_{\mathcal{K}_t} \leq 1}\left(\left(\frac{\delta_{t_{i'}t}}{\varphi_t(x_{i'})} - 1\right)\pi_t(x_{i'})f_t(x_{i'}) - \left(\frac{\delta_{t'_{i'}t}}{\varphi_t(x'_{i'})} - 1\right)\pi_t(x'_{i'})f_t(x'_{i'})\right) \leq \frac{2}{n}\alpha\Gamma.$$

By McDiarmid's inequality, $\mathbb{P}\left(\Xi_t(X_{1:n}, T_{1:n}) \geq \mathbb{E}[\Xi_t(X_{1:n}, T_{1:n})] + \eta\right) \leq e^{-n\eta^2\alpha^{-2}\Gamma^{-2}/2}$. Let $\xi_i(\pi_t, f_t) = \left(\frac{\delta_{T_i t}}{\varphi_t(X_i)} - 1\right)\pi_t(X_i)f_t(X_i)$ and note that for all $\pi_t, f_t$ we have $\mathbb{E}[\xi_i(\pi_t, f_t)] = \mathbb{E}[(\mathbb{E}[\delta_{T_i t}/\varphi_t(X_i) \mid X_i] - 1)\pi_t(X_i)f_t(X_i)] = 0$ and that $\xi_1(\cdot, \cdot), \xi_2(\cdot, \cdot), \ldots$ are iid. Therefore, letting $\xi'_1(\cdot, \cdot), \xi'_2(\cdot, \cdot), \ldots$ be iid replicates of $\xi_1(\cdot, \cdot), \xi_2(\cdot, \cdot), \ldots$ (ghost sample) and letting $\rho_i$ be iid Rademacher random variables independent of all else, we have

$$\mathbb{E}[\Xi_t(X_{1:n}, T_{1:n})] = \mathbb{E}[\sup_{\pi_t \in \Pi_t, \|f_t\|_{\mathcal{K}_t} \leq 1} \frac{1}{n}\sum_{i=1}^n (\mathbb{E}[\xi'_i(\pi_t, f_t)] - \xi_i(\pi_t, f_t))]$$

$$\leq \mathbb{E}[\sup_{\pi_t \in \Pi_t, \|f_t\|_{\mathcal{K}_t} \leq 1} \frac{1}{n}\sum_{i=1}^n (\xi'_i(\pi_t, f_t) - \xi_i(\pi_t, f_t))]$$

$$= \mathbb{E}[\sup_{\pi_t \in \Pi_t, \|f_t\|_{\mathcal{K}_t} \leq 1} \frac{1}{n}\sum_{i=1}^n \rho_i(\xi'_i(\pi_t, f_t) - \xi_i(\pi_t, f_t))]$$

$$\leq 2\mathbb{E}[\sup_{\pi_t \in \Pi_t, \|f_t\|_{\mathcal{K}_t} \leq 1} \frac{1}{n}\sum_{i=1}^n \rho_i\xi_i(\pi_t, f_t)].$$

Note that by bounded kernel we have $\|\mathcal{K}_t(x, \cdot)\|_{\mathcal{K}_t} = \sqrt{\mathcal{K}_t(x,x)} \leq \Gamma$ and therefore

$$\sup_{\|f_t\|_{\mathcal{K}_t} \leq 1, x \in \mathcal{X}} f_t(x) = \sup_{\|f_t\|_{\mathcal{K}_t} \leq 1, x \in \mathcal{X}} \langle f_t, \mathcal{K}_t(x, \cdot)\rangle \leq \sup_{\|f_t\|_{\mathcal{K}_t} \leq 1, \|g\|_{\mathcal{K}_t} \leq \Gamma} \langle f_t, g\rangle = \Gamma.$$

As before, $\|\xi_1 - \xi_2\|_{\mathcal{K}_t}^2 + \|\xi_1 + \xi_2\|_{\mathcal{K}_t}^2 = 2\|\xi_1\|_{\mathcal{K}_t}^2 + 2\|\xi_2\|_{\mathcal{K}_t}^2 + 2\langle\xi_1, \xi_2\rangle - 2\langle\xi_1, \xi_2\rangle = 2\|\xi_1\|_{\mathcal{K}_t}^2 + 2\|\xi_2\|_{\mathcal{K}_t}^2$ implies by induction that $\sum_{\rho_i \in \{-1, +1\}^n} \|\sum_{i=1}^n \rho_i\xi_i\|_{\mathcal{K}_t}^2 = 2^n\sum_{i=1}^n \|\xi_i\|_{\mathcal{K}_t}^2$. Hence,

$$\mathbb{E}[\|\frac{1}{n}\sum_{i=1}^n \rho_i E_{X_i}\|_{\mathcal{K}_t}] \leq (\mathbb{E}[\|\frac{1}{n}\sum_{i=1}^n \rho_i E_{X_i}\|_{\mathcal{K}_t}^2])^{1/2} = (\frac{1}{n^2}\sum_{i=1}^n \mathbb{E}[\|E_{X_i}\|_{\mathcal{K}_t}^2])^{1/2} \leq \Gamma/\sqrt{n}.$$

Note that $|\frac{\delta_{T_i t}}{\varphi_t(X_i)} - 1| \leq \alpha$, that $x^2$ is $2b$-Lipschitz on $[-b, b]$, and that $ab = \frac{1}{2}((a+b)^2 - a^2 - b^2)$. Therefore, by the Rademacher comparison lemma [28, Thm. 4.12], we have

$$\mathbb{E}[\Xi_t(X_{1:n}, T_{1:n})] \leq 2\alpha\mathbb{E}[\sup_{\pi_t \in \Pi_t, \|f_t\|_{\mathcal{K}_t} \leq 1} \frac{1}{n}\sum_{i=1}^n \rho_i\pi_t(X_i)f_t(X_i)]$$

$$\leq \alpha\mathbb{E}[\sup_{\pi_t \in \Pi_t, \|f_t\|_{\mathcal{K}_t} \leq 1} \frac{1}{n}\sum_{i=1}^n \rho_i(\pi_t(X_i) + f_t(X_i))^2]$$

$$+ \alpha\mathbb{E}[\sup_{\pi_t \in \Pi_t} \frac{1}{n}\sum_{i=1}^n \rho_i\pi_t(X_i)^2] + \alpha\mathbb{E}[\sup_{\|f_t\|_{\mathcal{K}_t} \leq 1} \frac{1}{n}\sum_{i=1}^n \rho_i f_t(X_i)^2]$$

$$\leq 4\Gamma\alpha\mathbb{E}[\sup_{\pi_t \in \Pi_t, \|f_t\|_{\mathcal{K}_t} \leq 1} \frac{1}{n}\sum_{i=1}^n \rho_i(\pi_t(X_i) + f_t(X_i))]$$

$$+ 2\alpha\mathbb{E}[\sup_{\pi_t \in \Pi_t} \frac{1}{n}\sum_{i=1}^n \rho_i\pi_t(X_i)] + 2\Gamma\alpha\mathbb{E}[\sup_{\|f_t\|_{\mathcal{K}_t} \leq 1} \frac{1}{n}\sum_{i=1}^n \rho_i f_t(X_i)]$$

$$\leq 6\Gamma\alpha(\mathfrak{R}_n(\Pi_t) + \Gamma/\sqrt{n}).$$

Next, let $\omega_{ti}(\pi_t) = (\delta_{T_i t}/\varphi_{T_i t} - 1)\pi_t(X_i)$ and $\Omega_t(X_{1:n}, T_{1:n}) = \sup_{\pi_t \in \Pi_t} \frac{1}{n}\sum_{i=1}^n \omega_{ti}(\pi_t)$. Note that $\sup_{\pi \in \Pi}(Z(\pi) - 1) \leq \sum_{t=1}^m \Omega_t(X_{1:n}, T_{1:n})$. Fix $x_{1:n}, t_{1:n}, x'_{1:n}, t'_{1:n}$ such that $x'_i = x_i, t'_i = t_i \, \forall i \neq i'$ and note that

$$\Omega_t(x_{1:n}, t_{1:n}) - \Omega_t(x'_{1:n}, t'_{1:n}) \leq \frac{1}{n}\sup_{\pi_t \in \Pi_t}\left(\left(\frac{\delta_{t_{i'}t}}{\varphi_t(x_{i'})} - 1\right)\pi_t(x_{i'}) - \left(\frac{\delta_{t'_{i'}t}}{\varphi_t(x'_{i'})} - 1\right)\pi_t(x'_{i'})\right) \leq \frac{2}{n}\alpha$$

By McDiarmid's inequality, $\mathbb{P}\left(\Omega_t(X_{1:n}, T_{1:n}) \geq \mathbb{E}[\Omega_t(X_{1:n}, T_{1:n})] + \eta\right) \leq e^{-n\eta^2\alpha^{-2}/2}$. Note that $\mathbb{E}[\omega_{ti}(\pi_t)] = 0$ for all $\pi_t$ and that $\omega_{t1}(\cdot), \omega_{t2}(\cdot), \ldots$ are iid. Using the same argument as before, letting $\rho_i$ be iid Rademacher random variables independent of all else, we have

$$\mathbb{E}[\Omega_t(X_{1:n}, T_{1:n})] \leq 2\mathbb{E}[\sup_{\pi_t \in \Pi_t} \tfrac{1}{n}\sum_{i=1}^n \rho_i \omega_{ti}(\pi_t)] \leq 2\alpha\mathfrak{R}_n(\Pi_t).$$

With a symmetric argument, letting $\delta = 3m\nu/(3m+2)$, with probability at least $1 - 2\delta/3$, we have $\sup_{\pi \in \Pi}|1 - Z(\pi)| \leq 2\alpha\mathfrak{R}_n(\Pi) + \alpha\sqrt{2\log(3m/\delta)/n} \leq 2\alpha\mathfrak{R}_n(\Pi) + \alpha\sqrt{2\log(4m/\nu)/n} \leq 1/2$.

Since $\|\tilde{W}\|_2 \leq \sqrt{n}\alpha/Z(\pi)$, we get that, with probability at least $1 - \delta$, both $\sup_{\pi \in \Pi}\|\tilde{W}\|_2 \leq 2\alpha\sqrt{n}$ and for all $t \in [m]$

$$\sup_{\pi \in \Pi}\mathfrak{B}_t(\tilde{W}, \pi_t; \|\cdot\|_{\mathcal{K}_t}) \leq \alpha\Gamma(12\mathfrak{R}_n(\Pi_t) + 2\mathfrak{R}_n(\Pi) + 12\Gamma/\sqrt{n} + 3\sqrt{2\log(3m/\delta)/n}).$$

Therefore, with probability at least $1 - \delta$, using twice that $\ell_1$ is the biggest $p$-norm,

$$\begin{aligned}
\mathcal{E} = \sup_{\pi \in \Pi}\mathfrak{E}(W_n^*, \pi; \|\cdot\|_{p,\mathcal{K}_{1:m},\gamma_{n,1:m}}, \Lambda_n) &\leq \sup_{\pi \in \Pi}\mathfrak{E}(\tilde{W}, \pi; \|\cdot\|_{p,\mathcal{K}_{1:m},\gamma_{n,1:m}}, \Lambda_n)\\
&\leq \sum_{t=1}^m \gamma_t \sup_{\pi \in \Pi}\mathfrak{B}_t(\tilde{W}, \pi_t; \|\cdot\|_{\mathcal{K}_t}) + \tfrac{\overline{\kappa}}{n}\sup_{\pi \in \Pi}\|\tilde{W}\|_2\\
&\leq 8\alpha\Gamma\overline{\gamma}m\mathfrak{R}_n(\Pi) + \tfrac{2\alpha\overline{\kappa} + 12\alpha\Gamma^2\overline{\gamma}m + 3\alpha\Gamma\overline{\gamma}m\sqrt{2\log(3m/\delta)}}{\sqrt{n}}.
\end{aligned}$$

Consider case (a). Note that $\sup_{\pi \in \Pi}\sum_{t=1}^m |B_t(W_n^*, \pi_t; \mu_t)| \leq \|\mu\|\,\mathcal{E}$ and $\sup_{\pi \in \Pi}\|W_n^*\|_2 \leq \underline{\kappa}^{-1}\mathcal{E}$. Since $\mathbb{E}[\sum_{i=1}^n W_i \epsilon_i \mid X_{1:n}, T_{1:n}] = 0$, $\epsilon_i \in [-B, B]$ and $W_i\epsilon_i' - W_i\epsilon_i'' \leq 2BW_i$ for $\epsilon_i', \epsilon_i'' \in [-B, B]$, by McDiarmid's inequality (conditional on $X_{1:n}, T_{1:n}$), we have that with probability at least $1 - \delta'$, $|\sum_{i=1}^n W_{ni}^*\epsilon_i| \leq \|W_n^*\|_2 B\sqrt{2\log(2/\delta)}$. Therefore, letting $\delta' = 2\nu/(3m+2)$ so that $3m/\delta = 2/\delta' = (3m+2)/\nu \leq 4m/\nu$, with probability at least $1 - \nu$, we have

$$\sup_{\pi \in \Pi}|\tau_{W_n^*} - \mathrm{SAPE}(\pi)| \leq 8\alpha\Gamma\overline{\gamma}m(\|\mu\| + \sqrt{2\log(4m/\nu)}\underline{\kappa}^{-1}B)\mathfrak{R}_n(\Pi)$$

$$+ \tfrac{2\alpha\overline{\kappa}\|\mu\| + 12\alpha\Gamma^2\overline{\gamma}m\|\mu\| + (2\alpha\overline{\kappa}\underline{\kappa}^{-1}B + 12\alpha\Gamma^2\overline{\gamma}m\underline{\kappa}^{-1}B + 3\alpha\Gamma\overline{\gamma}m\|\mu\|)\sqrt{2\log(4m/\nu)} + 6\alpha\Gamma\overline{\gamma}m\underline{\kappa}^{-1}B\log(4m/\nu)}{\sqrt{n}}.$$

This gives the first result in case (a). The second is given by noting that, by McDiarmid's inequality, with probability at least $1 - \nu/(4m)$, $\mathfrak{R}_n(\Pi_t) \leq \widehat{\mathfrak{R}}_n(\Pi_t) + 4\sqrt{2\log(4m/\nu)}$. Case (b) is given by following a similar argument as in the proof of Thm. 3(b). $\qquad\square$

*Proof of Cor. 7.* These results follow directly from the proof of Thm. 6, the convergence in particular of $\mathfrak{E}^2(W^*(\pi), \pi; \|\cdot\|, \Lambda)$, the decomposition of the DR estimator in Thm. 1, and a standard Rademacher complexity argument concentrating $\mathrm{SAPE}(\pi)$ uniformly around $\mathrm{PAPE}(\pi)$. $\qquad\square$

# B   IPW and DR weight SVM details

To reduce training a deterministic linear policy using IPW evaluation to weighted SVM classification, we add multiples of $\sum_{i=1}^n \pi_{T_i}(X_i)/\hat{\phi}_{T_i}(X_i)$ (1 in expectation) and note that

$$\begin{aligned}
\tfrac{1}{B}\left(\hat{\tau}^{\mathrm{IPW}}(\pi) - C\sum_{i=1}^n \tfrac{\pi_{T_i}(X_i)}{\hat{\phi}_{T_i}(X_i)} - \sum_{i=1}^n \tfrac{Y_i - C}{\hat{\phi}_{T_i}(X_i)}\right) &= \sum_{i=1}^n \tfrac{C - Y_i}{B\hat{\phi}_{T_i}(X_i)}(1 - \pi_{T_i}(X_i))\\
&= \sum_{i=1}^n \tfrac{C - Y_i}{B\hat{\phi}_{T_i}(X_i)}\mathbb{I}[T_i \neq \tilde{T}_{\pi}(X_i)].
\end{aligned}$$

Choosing $C$ sufficiently large so that all coefficients are nonnegative and choosing $B$ so that all coefficients are in $[0, 1]$, we replace the indicators $\mathbb{I}[T_i \neq \tilde{T}_{\pi(X_i)}]$ with their convex envelope hinges to come up with a weighted version of Crammer and Singer [12]'s multiclass SVM.

For the DR version, we replace $\pi_t(X_i)$ with $\mathbb{I}[t = \tilde{T}_{\pi(X_i)}]$ and we do the above with but using $\hat{\epsilon}_i$ and also add multiples of $\hat{\tau}^{\mathrm{direct}}(1(\cdot)) = \sum_{i=1}^n \sum_{t=1}^m \mu_t(X_i)$ to make all indicators be 0-1 loss and have nonnegative coefficients. Replacing indicators with hinge functions, we get a weighted multiclass SVM with different weights for each observation and *each error* type.