[Reviews · NeurIPS 2018]

Reviewer 1



The Author addresses the tasks of policy evaluation and learning from observational data by directly deriving weights that optimize for balance between the weighted data and the target policy. The proposed approach avoids the shortcomings of the widely-used inverse propensity weighting approach – namely, high variance and inefficient learning – and obtains, on a few toy examples, better results; theoretical consistency guarantees and regret bounds presumably underlie this improved performance. While the proposed approach seems novel, the paper is very condensed – both conceptually and textually – and hard to read. The method is said to be “computationally intensive” and, indeed, even using a commercial optimizer, the analyzed examples are very small, both in terms of the number of samples (up to a few hundreds) and features (<10), much smaller than typical real-world datasets, e.g., derived from electronic health records, the paper’s motivating example. Please report run times to demonstrate the extent of this challenge. Table 2, can you explain the better performance of Balanced vs Balanced-DR?

Reviewer 2



Summary: The paper proposes a new approach to policy learning and evaluation that uses weighted estimators (like most of the literature) but avoids the intermediate estimation of a propensity score -- instead, the authors propose to directly seek weights that minimize a certain notion of balance. They illustrate their methods with simulations, and offer theoretical guarantees. I believe that the paper has strong points. I appreciate the effort that the authors made to introduce the reader to this literature, by setting up the problem carefully and reviewing the literature in some details (Section 1.1 and Section 1.2). I believe that the method -- as I understand it -- is theoretically sensible. My criticism of the paper centers around two axes: the practical interest of the approach, and the readability of the exposition. Detailed comments follow. This reviewer did not check the proofs. Main comments: 1) I wish to reiterate my earlier appreciation of the effort made by the authors to write an accessible introduction to the policy evaluation and policy learning literature. Coming from a causal inference background, I found the review of existing approaches very useful and adequately detailed. 2) I also acknowledge the fact that the authors have endeavored to provide very general results. Theorem 1 and Corollary 2 deal with both weighted estimators and DR estimators. The following paragraph deals with general RKHS. Lemma 1 offers multiple interpretations for the CMSE objective. Section 3.2 explores consistency under a variety of assumptions. I recognize the effort that the authors have put into this paper. 3) My first serious criticism of the paper is its overall readability: A) Although it displays the efforts put by the authors, I believe that the generality of the paper hurts its readability. My impression is that the paper is trying to do too much, and the main results end up being obscured. For instance, I believe that dealing with both weighted estimators and doubly robust estimators is not helpful -- it would have been better, in my opinion, to spend more time and develop more intuition for the simple case, while referring the reader to the appendix for the DR results. It is unclear what Lemma 1 adds to the overall exposition of their method; it is interesting, but not crucial to the paper, and thus distracting. Presenting results and assumptions in terms of $C_0$ universal kernels seems unnecessarily general. It would have been better to focus on the most widely used kernel and then offer generalizations in appendix. I am not sure that the generality afforded by the use of the Rademacher complexity contributed much to the overall message of the paper. B) One consequence of this is that it is sometimes difficult to see what the flow of the paper is, and what each subsection is trying to accomplish. What is the purpose of Lemma~1 in the overall picture? The connection with Causal Inference is not crucial, and distracts from the main point of the paper. Is it necessary to have the results of both Theorem 3 and Corollary 4? Is it necessary to have both Theorem~6 and Corollary~7? Etc... C) The general impression is that the true core of the paper is small -- contained in Theorem~1, and lines 150-152. The rest is an attempt to justify the method by providing examples and theoretical results, neither of which I found of practical interest -- more on that later. D)Trying to cram so many different results in 10 pages made the layout of the paper very, very dense and, in places, unreadable. Theorem 3 and Corollary 4 are examples. Most equations are inlined. There is no whitespace before or after definitions, theorems, etc... 4) My second criticism concerns the practical interest of the paper. A) I believe that theoretical guarantees are important. But serious effort should be made to assess whether the assumptions made are practically relevant and are likely to hold in practice, etc... This is another place where the generality of the results is problematic. Section~3.2 is a particularly stark example of that: what are the actual implications of the assumptions and results of Section~3.2? To be fair, the paragraph after Corollary 4 (lines 211-217) is trying to do that, but I think it is somewhat under-developed. B) Another example in which the practicality of the method is obscured is with the use of the logit parametrized policy class. In Example 3 they generate data from a multilogit model (thus, compatible with the parametrized policy class). But what if the data generating process is not even remotely close to being a logit? What if it is not continuous in $x$? C) The authors hint at it in the conclusion, but a longer discussion of the computational complexity would have been useful. Including even a rough idea of the execution time in Table~1 would have been helpful. D) In general, I would have liked the examples to be more developed, explore misspecification in more details, etc... A few minor comments: 1) You criticize the two-stanatureoftheapproachestheliterature,andthenoseabilevel' approach in Section~3. What is the difference between two-staandbilevel'. 2) l. 22-23: transform something oratherthanto'. 3) l. 81: '[...] the estimator by via' should be either byorvia'. 4) l. 124: the sentence `Minimal CMSE is the target of choosing weights for weighting-based policy evaluation' is confusing.

Reviewer 3



The authors proposed a new approach to policy evaluation and learning in causal inference. Traditional approached to this problem have used either regression-based techniques, propensity score weighting (i.e., the probability of receiving a particular treatment at a fixed level of the covariates) or so-called doubly robust methods (that use both weighting and regression). The paper proposes instead of using weights inversely proportional to the propensity score to use balance weights which are computed directly by optimizing for balance between the weighted data and the target policy. The authors then derive some theoretical guarantees based on their proposed methodology. The key insight of this paper is to recognize that the weighting can be computed directly rather than relying on the propensity score. I think this research is closely connected with the work done by Jose Zubizarreta (citation given at the end of this). Although the application is different, the key insight of using weights that are directly optimized is similar to Zubizarreta's work. It would be good if these connections were discussed further, in particular, what are the similarities and differences (besides the application)? The technical work is excellent, although at times a little challenging to follow. This is mostly due to the amount of information that is packed in the paper. Using simpler and clearer examples could help to alleviate some of this. For example, Example 1 is not a conventional example that I am familiar with, I wonder if using something closer to a real-world application would help this? I would also like to see some more discussion regarding the different assumptions that are made. For example, what happens if strong overlap does not hold? Zubizarreta, J. R. (2015), "Stable Weights that Balance Covariates for Estimation with Incomplete Outcome Data,"